# What Are the Best Pollinator Candidates for *Camelia oleifera*: Do Not Forget Hoverflies and Flies

**DOI:** 10.3390/insects13060539

**Published:** 2022-06-11

**Authors:** Bin Yuan, Guan-Xing Hu, Xiao-Xiao Zhang, Jing-Kun Yuan, Xiao-Ming Fan, De-Yi Yuan

**Affiliations:** 1Key Laboratory of Cultivation and Protection for Non-Wood Forest Trees, Ministry of Education, Central South University of Forestry and Technology, Changsha 410004, China; yuan_bin322@163.com (B.Y.); csuft4880@gmail.com (G.-X.H.); hymlrzxx@163.com (X.-X.Z.); yuanjingku0830@163.com (J.-K.Y.); 2Key Lab of Non-Wood Forest Products of State Forestry Administration, Central South University of Forestry and Technology, Changsha 410004, China

**Keywords:** *Camellia oleifera* Abel., pollinator candidates, *Apis mellifera*, hoverflies, flies

## Abstract

**Simple Summary:**

*Camellia oleifera* is an important woody grain and oil plant worldwide. However, owing to a significant decline in the number of wild pollinators globally and the associated reduction in pollination services and seed production, camellia oil is in short supply. Therefore, it is important to evaluate the pollination mechanisms and efficiency of wild pollinators in *C. oleifera* seed production. We explored the pollination system of *C. oleifera*, focusing on the flower-visiting characteristics of its candidate pollinators. We found that *Apis mellifera* is the best candidate pollinator, but flies and hoverflies also play important roles in the pollination system.

**Abstract:**

*Camellia oleifera* Abel. is an important woody oil plant, and its pollination success is essential for oil production. We conducted this study to select the best pollinator candidates for *C. oleifera* using principal component analysis and multi-attribute decision-making. Field observations of the flower-visiting characteristics of candidate pollinators were conducted at three sites. The insect species that visited flowers did not considerably differ between regions or time periods. However, the proportion of each species recorded did vary. We recorded eleven main candidates from two orders and six families at the three sites. The pollen amount carried by *Apis mellifera* was significantly higher than that of other insects. However, the visit frequency and body length of *Apis mellifera* were smaller than those of *Vespa velutina*. Statistical analysis showed that *A. mellifera* is the best candidate pollinator; *Eristalis*
*cerealis* is a good candidate pollinator; *Phytomia zonata*, *A. cerana,* and *V. velutina* were ordinary candidate pollinators; and four fly species, *Episyrphus balteatus*, and *Eristalinus arvorum* were classified as inefficient candidate pollinators. Our study shows that flies and hoverflies play an important role in the pollination system. Given the global decline in bee populations, the role of flies should also be considered in *C. oleifera* seed production.

## 1. Introduction

Many plants have mutualistic relationships with pollinators that ensure their reproductive success [1]. Pollinators also help maintain the genetic diversity of plant populations [2,3,4]. However, these benefits come at a cost, as pollinators consume nectar, pollen, pseudopollen, and other flower resources in return for maintaining the plant numbers [5]. The mutually beneficial relationship between plants and their pollinators plays a key role in maintaining ecosystem stability [6]. However, the number and diversity of pollinators have declined globally owing to habitat fragmentation, pesticide use, climate change, and other factors. During the 1850s in the United Kingdom, 23 bee and wasp species were declared to be on the verge of extinction [7]. Subsequently, plant reproduction has also decreased. In the context of cash crops, a decline in the variety and number of pollinators can reduce their yields by 5–8%, reaching levels close to extinction for some species. However, the level of extinction risk from pollinator decline depends on the extent to which a species relies on its pollinators for reproduction [8]. Understanding the pollination efficiency of pollinators can help us support these interactions and predict the risk of plant extinction in the context of declining pollinator populations [9].

Insects have been identified as the world’s most important pollinators, contributing to 87% of pollination globally, making their services critical for the sustainability of natural ecosystems [10,11]. *Camellia oleifera* Abel. is a woody, oil-producing plant native to China [12]. Its seeds are rich in unsaturated fatty acids and are used to produce edible tea oil (camellia oil) [13]. *C. oleifera* is a self-incompatible plant that relies heavily on active insect pollination [14]. When it blossoms, *C. oleifera* attracts more than 50 species of pollinating insects, including bees, wasps, hoverflies, and flies [15]. However, managed honeybee populations placed in *C. oleifera* forests have a high mortality rate because *C. oleifera* nectar contains strong alkaloids and other indigestible compounds that cause posterior intestinal obstruction in bee larvae [16,17]. Therefore, many studies have proposed the use of wild pollinators for *C. oleifera* pollination. However, the number of wild pollinators has been declining considerably worldwide, and the consequent reduction in pollination and seed production has caused a shortage in the camellia oil supply [8]. Therefore, it is important to identify candidate wild insect pollinators for *C. oleifera*.

In this study, we explored the pollination system of *C. oleifera* by observing the insect species that visited the flowers at different times across a range of different sites. The proportion of each insect species, flower visit duration and frequency, and other indices were quantified, and the body and flower-visiting characteristics of the insects were analysed and compared. The pollination potential of flower visitors was compared using principal component analysis and a multi-attribute decision-making model. Functional groups were determined on the basis of the scores obtained using cluster analysis.

## 2. Materials and Methods

### 2.1. Study Species and Sites

Our research was conducted in December 2021 in Yuelu District, Tianxin District, and Wangcheng District of Changsha, Hunan Province, China (Table 1).

### 2.2. Identification of Flower Visitors

To identify the visiting insects, we photographed those that sat on the *C. oleifera* flowers and used a sweep net for capture. The insects captured were stored in 10 mL vials containing 75% alcohol and were then transported to the laboratory for identification by the relevant experts [18].

### 2.3. Flower-Visiting Density

To select the best observation period for the flower-visiting insects, the flower-visiting density of *C. oleifera* was recorded and classified from 8:00 to 18:00 on three sunny days when the shrubs were in full bloom. The statistical method for examining the flower-visiting density was as follows. Three rows of plants were randomly selected for each sampling session at the site, and then banded sampling was carried out. The number of insects landing on 200 flowers in the row was recorded back and forth for one hour, and the data from the three rows were added together to represent the overall flower-visiting density for the site. This experiment was carried out at three sites at the same time. After obtaining the data of each site, the average flower-visiting density in different periods at the three sites was recorded as data.

### 2.4. Insect Proportion

To quantify visits by pollinators to *C. oleifera* during the flowering period from November to January, we conducted the following experiments in December. The temperature in the first half of the month was 6–16 °C, and it was 3–11 °C for the remainder of the month. We observed a total of 2679 insects during the study. (a) We chose five sunny days during the first half of December at the three sites. Flower-visiting insects were observed from 11:00 a.m. to 12:00 p.m., which is when they are the most active [19]. (b) We observed the visits to each site during the second half of December. The proportion of visiting insect species was calculated using the following formula:Insect proportion = Single species insect number/total insect number

### 2.5. Body Measurements of the Main Pollinator Candidates

For comparative analysis of the body characteristics of the flower visitors, we used vernier callipers to measure the body length, the head width and length, and the back plate width and length of the visitor specimens collected, with the measurements repeated ten times for each species. We also photographed the main flower visitors using a stereo microscope (OLYMPUS-SZX16, Tokyo, Japan).

### 2.6. Foraging Behaviours

To explore behavioural differences among the pollinator candidates, we recorded the visiting time, namely the duration from the insect landing on the flower to the insect leaving. A total of 205 insects were recorded during this process. We also recorded the visiting frequency, namely the number of insect visits to the flowers in one minute. A total of 168 insects were recorded. Data on any foraging behaviours observed were collected [20].

### 2.7. Pollen Load Analysis

To quantify the pollen load capacity of the pollinator candidates, we added a small amount of detergent to the visitor samples stored in the 10 mL vials to remove any pollen particles carried by them [21]. We transferred 10 μL of the solution to a blood cell counting plate and counted the pollen grains, with the tests repeated at least ten times for each species. The numbers of normal pollen and pseudopollen were quantified separately using a microscope (OLYMPUS-BX51, Tokyo, Japan). All the bees used for the test had their pollen pellets removed beforehand. The pollen load was calculated as follows:(Counted pollen number × 1000)/number of insects in each bottle

### 2.8. Data Analyses

#### 2.8.1. One-Way Analyses of Variance

To explore the differences between the body and visit characteristics of the insects, the visiting time, frequency, pollen load, posture characteristics, and the canopies visited were analysed. All the analyses were carried out using SPSS, version 26 (SPSS Inc., Chicago, IL, USA), with the statistical significance set at *p* < 0.05.

#### 2.8.2. Correlation Analysis

To screen the factors for follow-up analysis, we analysed the correlation of the insect pollen load, the body surface characteristics, the body length, proportion, flower-visiting frequency, and the flower-visiting time using SPSS, version 26, with irrelevant factors excluded [22].

#### 2.8.3. Principal Component Analysis

To determine the best pollinator candidate, the score for each insect was obtained using principal component analysis (PCA). The data were standardised to a mean of zero and a variance of one before conducting the PCA, examining the contribution of each feature to different principal components. According to Gutten’s lower bound principle, eigenvalues of <1 were excluded [23].

#### 2.8.4. Determination of Multiple Attributes

To rank the insects as pollinator candidates, we established a score matrix according to the observations. The score for each feature was then determined using MATLAB 2019, and the score for each insect species was then obtained [24].

#### 2.8.5. Cluster Analysis

To determine the pollination function group, the insects were clustered after the score for each species had been obtained. The pollination function groups were classified using SPSS, version 26, with the intergroup connection method, and the distance between the pollinator candidates was set as the Euclidean distance [25].

## 3. Results

### 3.1. The Diversity and Changes in the Insects Visiting C. oleifera

From 8:00 to 18:00, four categories of flower-visiting insects were recorded visiting *C. oleifera*, and the main insect species were found to be the same during this period (Figure 1 and Figure 2). The most active periods for visits to the flowers were from 11:00 to 12:00. Therefore, observations were carried out from 11:00 to 12:00 to better show the characteristics of the main candidates during the most active period. Twelve species of insects were recorded, and they made contact with the flower stigmas at the three sites (Figure 2). The insects recorded during the pollinator observations were from two orders and six families. The proportion of species at the three sites was different; for example, *Vespa mandarinia* was not recorded at site 1 (Table 2).

*Apis mellifera* accounted for 66.06% of the *C. oleifera* visitors at site 3 but represented 3.57% of the visiting pollinators at site 1. The status of the flower visitors at the three sites was also different. At site 1, *V. velutina* represented the largest proportion of the species, but at sites 2 and 3, *A. mellifera* accounted for the largest proportion of insect flower visitors. Elastic regulation was also observed with time. During the first half of December, wasps were the main visiting insects, but in the second half of December, the proportions of wasps and bees fluctuated considerably. Flies and hoverflies formed a large proportion of the insect visitors (Table 3).

### 3.2. Daily Activity of Insects Visiting C. oleifera

The duration of visits varied considerably during the pollinator observations. For example, the longest visiting time for *Phytomia zonata* was 90 times the shortest recorded time. There was also a significant difference in the visit frequency (*p <* 0.05) of the insects. *V. velutina* visited flowers with the highest frequency of 4.5 times per minute, whereas *A. mellifera* and *Lucilia sericata* had slightly lower frequencies. There was no difference in the mean frequencies of *A. cerana* and some of the visiting hoverflies, and the mean frequency was the lowest for the flies (Table 4).

There were significant differences in the behaviours of different insects. Flies spent more than half of their time visiting the anthers and inhabited the anther instead of the stigma. Among the hoverflies, most showed foraging behaviours and frequently made contact with the stigmas. More than half of the hoverflies came into contact with the stigma during the observation process, except *Episyrphus balteatus,* which often hovered over but did not visit the flowers. Most of the bees visited to collect the nectar, but *A. mellifera* showed a higher enthusiasm and actively collected pollen on their pollen-carrying legs. The wasps were highly active, usually buried their heads in the anthers to forage, and made contact with the stigmas almost every time (Table 4).

There were significant differences in the powder-carrying capacity of the pollinator candidates (*p <* 0.05, Table 5). *A. mellifera* carried the largest amount, with more than 50,000 grains per individual, followed by *E. cerealis*, with more than 48,800 grains per individual. There were no significant differences for *Phytomia zonata*, *A. cerana,* and *V. velutina*. There were also significant differences in the proportion of normal pollen and pseudopollen carried by the different insects. The ratio of pseudopollen to the normal pollen carried by most insects was approximately 0.1, and the ratio of flies was generally large, among which *L. sericata* had the highest ratio of up to 0.43.

### 3.3. Body Measurements of the Main Visiting Insects

Among the 11 main pollinator candidates, *V. velutina* was the largest species, with its body length reaching 20.42 mm. Bees and hoverflies were similar in appearance, but the bees were larger in size (*p <* 0.05, Table 6).

There were also many differences in the body surface characteristics among these insects. Considering the proportion of the species, the visiting time, frequency, and the pollen load, we selected *E. cerealis*, *A. mellifera*, and *V. velutina* for further analysis. *E. cerealis* is densely tomentose, especially on the back. *A. mellifera* has dense yellow villi on the body and dense hairs on the feet. *V. velutina* has dense villi on the body surface, especially on the chest. The flies are mainly covered with relatively hard and smooth bristles and short black hair. Among the hoverflies, *P. zonata* has dense villi on the body and feet, and the pollen grains could be seen on the specimens. *E. arvorum* is fluffy on the back, while *E. balteatus* is fluffy on both sides of the body (Figure 3).

### 3.4. Determination of the Best Pollinator Candidate

Using correlation analysis, we selected five factors with high correlation from the six previously recorded factors for follow-up analysis (*p* < 0.01, Table 7). PCA with eigenvalues of >1 contributed to 82.66% of the total cumulative variance (Table 8). Then, we calculated the comprehensive scores for 11 species. Only the top five insects had positive scores (Table 9). In the multi-attribute decision-making (MADM) model, the values for the decision matrix and the weight of five variables of the dominant flower visitors were calculated by examining the influence of each factor on the pollinator candidates (Table 10). We obtained the scores for each insect species. The score for *A. mellifera* was much higher than that of the other insects, reaching 0.91 (Table 11).

Two kinds of clustering results were obtained according to different methods. The two results showed that *A. mellifera* is the best pollinator candidate; *E. cerealis* is a ‘good’ pollinator candidate; *P. zonata*, *A. cerana*, and *V. velutina* are ‘ordinary’ pollinator candidates; four fly species, *E. balteatus,* and *E. arvorum* were classified as ‘inefficient’ pollinator candidates (Figure 4).

## 4. Discussion

Considering the body and flower-visiting characteristics of the candidate pollinators, bees might be the best candidate pollinators. However, flies and hoverflies also play an important role in *C. oleifera* pollination [26,27]. When land-use change, climate change, habitat fragmentation, and alien biological invasions occur, the diversity of insect pollinators in the ecosystem also changes. Insect pollinator communities can leverage functional redundancy to ensure the stability of an ecosystem through pollination function on a small spatial scale [28,29]. For example, when the chemical in *C. oleifera* nectar results in a reduction in the bee numbers, other pollinators, such as flies and hoverflies, become the dominant pollinators and compensate for the absence of the bees [30]. This dynamic regulation occurs because there is niche overlap among pollinator functional groups (species) [31]. This shows that an ecosystem with abundant pollinator functional groups can easily compensate for the loss of one or more groups [6].

In line with a previous study [15], we found that unlike bees and wasps, which are highly motivated to visit flowers and touch the stigma almost every time they visit flowers, flies are small in size, have a low pollen load, and are not highly motivated to visit flowers, often perching on the anthers without making contact with the stigma, which makes it difficult for them to pollinate the plants. However, they regulate the stability of the pollination system as generalised pollinators. In some plant species, flies also act as specialised pollinators [32,33]. Unlike flies, hoverflies have a high pollen load, and most individuals make contact with the stigma when they eat nectar, thereby transferring the pollen to the stigma and causing pollination. Hoverflies can increase the crop yield to the same extent as bees. Furthermore, hoverflies have certain advantages over bees, such as the ability to carry pollen further, thus promoting the movement of pollen and the flow of genes throughout the landscape [34,35]. Therefore, attention should be paid to the pollination efficiency of flies and hoverflies considering the gradual decline in bee populations [36].

We used statistical methods to select *A. mellifera* as the best *C. oleifera* candidate pollinator and found that flies and hoverflies also play crucial roles in the pollination system. PCA and the MADM model showed similar results for the pollinator potential of flower visitors. However, the results of both analyses differed from those based on taxonomic affinities. This suggests that the classification of flower visitors into different functional groups should consider the contribution of different flower visitors to pollination while disregarding the taxonomic affinities [37,38]. Like most flowering plants, *C. oleifera* has multiple visitor functional groups, including flies, hoverflies, bees, and wasps. However, only a subset of these visitors serves as effective pollinators [39]. Most studies adhere to traditional or taxonomic groups to classify functional groups. Taxonomically related insects show many differences with respect to posture, visiting enthusiasm, distribution, and plant–pollinator interactions [38,40].

Additionally, we determined that *E. cerealis* is a ‘good’ candidate pollinator; *P. zonata*, *A. cerana*, and *V. velutina* are ‘ordinary’ candidate pollinators; and the remaining flower visitors are ‘inefficient’ candidate pollinators for *C. oleifera*. This division highlights the relative importance of diverse pollinators [38]. In this study, the most important indicator of the pollinator potential of flower visitors was the pollen load. However, unlike most plants, *C. oleifera* has pseudopollen, which is ineffective for pollination [41]. Therefore, we distinguished between the pollen and the pseudopollen to better understand the effectiveness of pollen carried by insects. The results showed that pseudopollen may not affect pollination by insects. Although insects carry pseudopollen during flower visits, the proportion of pseudopollen was approximately 0.1, which is lower than the pseudopollen-to-pollen ratio of 0.3 on the *C. oleifera* anthers [42]. This indicates that the collection of pseudopollen rather than pollen by insects may be selective. However, the criteria for this selection are not clear.

## Figures and Tables

**Figure 1 insects-13-00539-f001:**
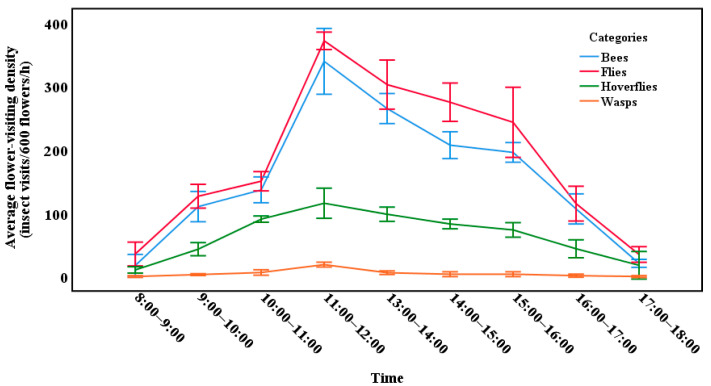
Density of pollinators visiting the flowers of *Camellia oleifera*.

**Figure 2 insects-13-00539-f002:**
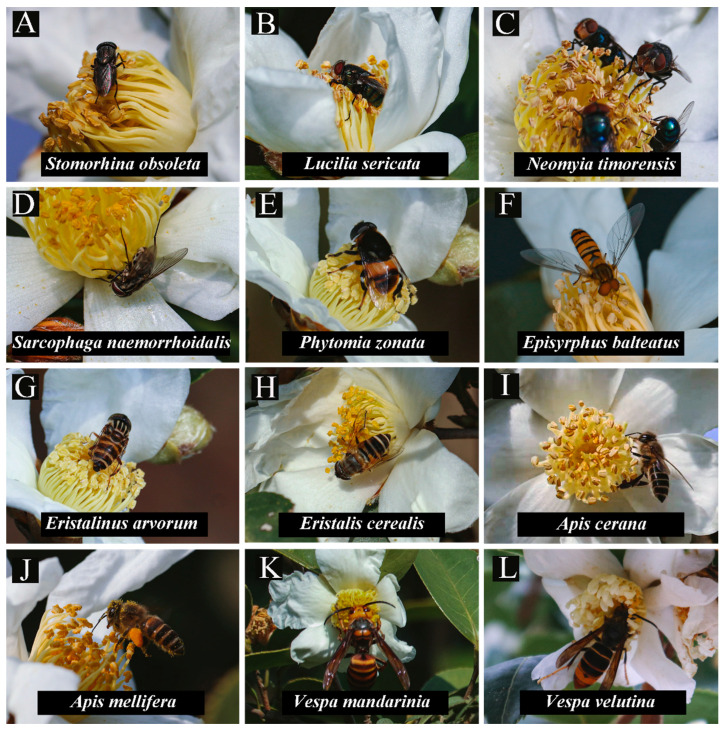
Insect flower visitors for *Camellia oleifera*.

**Figure 3 insects-13-00539-f003:**
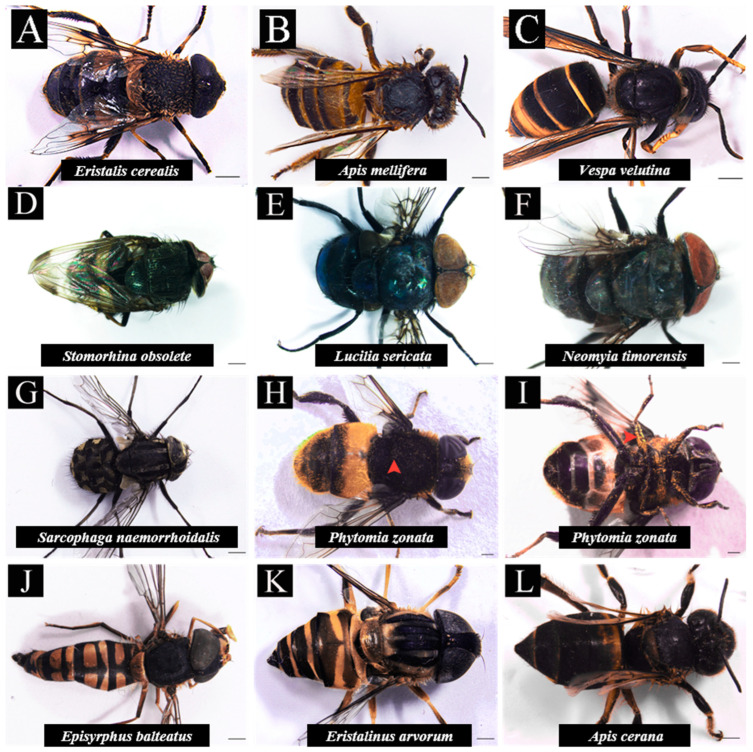
Posture of *Camellia oleifera* flower visitors. The red arrow represents the remaining pollen on the surface of the insect after taking it out of the penicillin bottle. Bar = 1 mm.

**Figure 4 insects-13-00539-f004:**
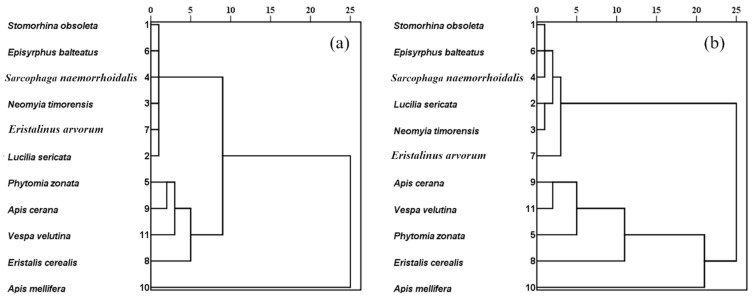
Cluster analysis of two methods. (**a**) principal component analysis. (**b**) Multi-attribute decision-making.

**Table 1 insects-13-00539-t001:** Distribution of *Camellia oleifera* at the study sites in Changsha, China, and the sampling methods employed for data collected at different sites.

No.	Location	Latitude and Longitude	Basic Situation	Sampling Methods
1	Tianxin District	28°8′14″ N, 112°59′08″ E	Only small trees, many varieties, well-managed.	Trees in the east, west, south, north, and middle of the areas were selected.
2	Yuelu District	28°13′48″ N, 112°55′53″ E	Neat arrangement of trees but low planting density. Predominance of young *C. oleifera* trees.	One row each in the upper, middle, and lower parts of the terrace was selected; trees in the left, right, and centre of each row were targeted.
3	Wangcheng District	28°22′12″ N, 112°49′12″ E	Covers a large area; located far away from urban areas; many tall trees.	Eight plots of 10 m × 10 m were set up; trees in the east, west, south, north, and middle were selected.

**Table 2 insects-13-00539-t002:** Abundance of flower visitors to *Camellia oleifera* from three sites during the first half of December.

NO.	Order	Family	Species	Proportion		
				Site 1	Site 2	Site 3
A	Diptera	Calliphoridae	*Stomorhina obsoleta*	12.14%	9.74%	0.19%
B	Diptera	Calliphoridae	*Lucilia sericata*	5.00%	4.62%	1.81%
C	Diptera	Muscidae	*Neomyia timorensis*	4.29%	12.31%	8.53%
D	Diptera	Sarcophagidae	*Sarcophagan aemorrhoidalis*	11.43%	4.10%	0.56%
E	Diptera	Syrphidae	*Phytomia zonata*	11.43%	8.72%	1.72%
F	Diptera	Syrphidae	*Episyrphus balteatus*	1.43%	6.67%	8.72%
G	Diptera	Syrphidae	*Eristalinus arvorum*	8.57%	9.74%	1.39%
H	Diptera	Syrphidae	*Eristalis cerealis*	2.86%	3.08%	5.98%
I	Hymenoptera	Apidae	*Apis cerana*	2.14%	5.13%	3.20%
J	Hymenoptera	Apidae	*Apis mellifera*	3.57%	33.33%	66.06%
K	Hymenoptera	Vespidae	*Vespa mandarinia*	0.00%	1.03%	0.46%
L	Hymenoptera	Vespidae	*Vespa velutina*	37.14%	1.54%	1.39%

**Table 3 insects-13-00539-t003:** The proportion of insect flower visitors to *Camellia oleifera* during different periods at site 1.

Categories	Family	Species	Proportion		
			First Half of December	Second Half of December	Volatility
Flies	Calliphoridae	*Stomorhina obsoleta*	12.14%	20.60%	8.46%
		*Lucilia sericata*	5.00%	1.01%	−3.99%
	Muscidae	*Neomyia timorensis*	4.29%	8.04%	3.75%
	Sarcophagidae	*Sarcophaga naemorrhoidalis*	11.43%	4.52%	−6.91%
Hoverflies	Syrphidae	*Phytomia zonata*	11.43%	15.08%	3.65%
		*Episyrphus balteatus*	1.43%	10.05%	8.62%
		*Eristalinus arvorum*	8.57%	4.52%	−4.05%
		*Eristalis cerealis*	2.86%	3.02%	0.16%
Bees	Apidae	*Apis cerana*	2.14%	4.02%	1.88%
		*Apis mellifera*	3.57%	22.61%	19.04%
Wasps	Vespidae	*Vespa velutina*	37.14%	6.53%	−30.61%

**Table 4 insects-13-00539-t004:** The main foraging behaviours of flower visitors to *Camellia oleifera*.

Categories	Species	Time Visiting Each Flower		Single Flower Visit Frequency (times/min)			Main Foraging Behaviours
		Shortest	Longest	Lowest	Highest	Average Frequency	
Flies	*Stomorhina obsoleta*	9	96	1	5	2.17 ± 1.24 c	Spent more than half of their visit on the anthers but touched the stigma less.
	*Lucilia sericata*	16	22	2	4	3.00 ± 0.77 abc	
	*Neomyia timorensis*	8	160	1	5	2.25 ± 1.16 c	
	*Sarcophaga naemorrhoidalis*	8	54	1	6	2.69 ± 1.38 bc	
Hoverflies	*Phytomia zonata*	3	270	1	5	2.71 ± 1.28 bc	More active, and more than half of them touched the stigma.
	*Episyrphus balteatus*	7	21	1	2	1.33 ± 0.47 c	Mainly inspected the flowers and stayed for a short time.
	*Eristalinus arvorum*	5	78	1	6	2.71 ± 1.28 bc	Sometimes rested on flowers temporarily without any activity.
	*Eristalis cerealis*	5	34	1	3	2.67 ± 0.60 bc	Took a short time to forage but touched the stigma almost every time.
Bees	*Apis cerana*	3	46	1	3	2.33 ± 0.58 bc	Visiting time was short, and the enthusiasm for visiting flowers is low.
	*Apis mellifera*	2	148	1	7	4.06 ± 1.61 ab	Frequent flower visits; actively collected pollen and made contact with the stigma almost every time.
Wasps	*Vespa velutina*	2	127	1	6	4.47 ± 1.27 a	Actively visited the flower and touched the stigma almost every time.

Within columns, different letters indicate significant differences at *p* < 0.05.

**Table 5 insects-13-00539-t005:** Pollen-carrying situation of main flower visitors in *Camellia oleifera*.

Categories	Species	Pollen Load (Grain/Individual)				Main Powder-Carrying Position
		Total Amount of Pollen	Normal Pollen	Pseudopollen	Pseudopollen/Normal Pollen	
Flies	*Stomorhina obsoleta*	2320 ± 129.84 d	1700 ± 169.97 c	620 ± 220.10 c	0.38 ± 0.16 ab	Back plate and head
	*Lucilia sericata*	1600 ± 1074.97 d	1200 ± 1032.80 c	400 ± 699.21 c	0.43 ± 0.79 a	
	*Neomyia timorensis*	2000 ± 608.58 d	1833 ± 593.17 c	167 ± 175.68 c	0.10 ± 0.11 cd	
	*Sarcophaga naemorrhoidalis*	3188 ± 400.64 d	2688 ± 512.62 c	500 ± 238.37 c	0.21 ± 0.16 abcd	
Hoverflies	*Phytomia zonata*	12,667 ± 237.17 c	11,333 ± 210.82 b	1500 ± 241.96 b	0.12 ± 0.09 cd	Body surface and feet
	*Episyrphus balteatus*	733 ± 262.94 d	600 ± 262.94 c	133 ± 172.13 c	0.30 ± 0.42 abc	Back plate and head
	*Eristalinus arvorum*	3533 ± 688.53 d	3400 ± 733.67 c	133 ± 172.13 c	0.04 ± 0.06 d	Body surface
	*Eristalis cerealis*	48,800 ± 5391.35 b	48,200 ± 5202.56 a	667 ± 699.21 c	0.01 ± 0.02 d	Villi on the body surface
Bees	*Apis cerana*	14,650 ± 1106.80 c	13,300 ± 948.68 b	1350 ± 411.64 b	0.10 ± 0.03 cd	Pollen-carrying legs and villi
	*Apis mellifera*	55,167 ± 6549.81 a	47,583 ± 6120.09 a	7500 ± 707.11 a	0.16 ± 0.02 bcd	
Wasps	*Vespa velutina*	13,000 ± 4216.37 c	11,675 ± 4323.79 b	1325 ± 373.61 b	0.13 ± 0.05 cd	Villi on the body surface

Within columns, different letters indicate significant differences at *p* < 0.05.

**Table 6 insects-13-00539-t006:** Posture characteristics of the main flower-visiting insects for *Camellia oleifera*.

Categories	Species	Body Length (mm)	Head Width (mm)	Head Length (mm)	Shoulder Length (mm)	Shoulder Width (mm)	Body Surface Characteristics
Flies	*Stomorhina obsoleta*	6.95 ± 1.17 f	2.28 ± 0.51 fg	2.01 ± 0.43 ef	2.47 ± 0.58 f	2.17 ± 0.56 h	Body surface bristles
	*Lucilia sericata*	8.99 ± 0.88 e	3.44 ± 0.33 e	2.18 ± 0.24 def	3.62 ± 0.37 de	3.44 ± 0.42 cd	
	*Neomyia timorensis*	9.63 ± 0.95 de	3.58 ± 0.24 de	2.46 ± 0.60 cde	3.65 ± 0.41 de	3.25 ± 0.26 de	
	*Sarcophaga naemorrhoidalis*	6.60 ± 0.45 f	2.19 ± 0.35 g	1.85 ± 0.42 f	2.66 ± 0.21 f	2.77 ± 0.36 fg	Bristled, sparse at the back
Hoverflies	*Phytomia zonata*	13.49 ± 0.60 b	4.99 ± 0.18 b	2.81 ± 0.31 c	4.82 ± 0.41 b	5.07 ± 0.36 b	Densely tomentose
	*Episyrphus balteatus*	9.39 ± 0.63 e	2.57 ± 0.19 f	2.68 ± 0.41 cd	2.61 ± 0.14 f	2.43 ± 0.10 gh	Tomentose on both sides and short hairs on the ventral segment
	*Eristalinus arvorum*	10.63 ± 1.06 d	3.88 ± 0.14 cd	3.57 ± 0.12 b	3.98 ± 0.27 cd	3.77 ± 0.38 c	Dorsal plate is tomentose, and the ventral segment is short-haired
	*Eristalis cerealis*	12.58 ± 1.65 bc	4.15 ± 0.91 c	2.76 ± 0.32 c	4.12 ± 0.56 c	3.59 ± 1.03 cd	Densely tomentose, and the dorsal plate is particularly dense
Bees	*Apis cerana*	12.20 ± 0.73 c	3.66 ± 0.18 de	2.91 ± 0.72 c	3.70 ± 0.31 de	3.04 ± 0.33 ef	Densely covered with yellow villi, short ventral hairs; pollen-carrying legs
	*Apis mellifera*	13.05 ± 0.65 bc	3.45 ± 0.24 e	2.56 ± 0.71 cd	3.54 ± 0.34 e	2.97 ± 0.57 ef	
Wasps	*Vespa velutina*	20.42 ± 2.48 a	5.34 ± 0.26 a	4.33 ± 1.01 a	6.06 ± 0.47 a	6.67 ± 0.51 a	Densely tomentose

Within columns, different letters indicate significant differences at *p* < 0.05.

**Table 7 insects-13-00539-t007:** Correlation analysis with six screening variables for flower visitors of *Camellia oleifera*.

	Pollen Load	Body Surface Characteristics	Body Length	Proportion	Visiting Frequency	Visiting Time
Pollen load	1.00					
Body surface characteristics	0.74 **	1.00				
Body length	0.39	0.67	1.00			
Proportion	0.70	0.47	0.16	1.00		
Visiting frequency	0.46	0.51	0.70	0.43	1.00	
Visiting time	−0.53	−0.50	−0.32	−0.37	−0.20	1.00

** T-test for all variables, *p* < 0.01.

**Table 8 insects-13-00539-t008:** Score coefficient, variance contribution, and cumulative contribution rate of the two principal components.

Principal Component	Pollen Load	Body Surface Characteristics	Body Length	Proportion	Visiting Frequency	Eigenvalue	Variance Contribution	Cumulative Contribution Rates
PC1	0.27	0.28	0.24	0.22	0.25	3.11	62.19%	62.19%
PC2	0.38	−0.04	−0.60	0.61	−0.32	1.02	20.47%	82.66%

**Table 9 insects-13-00539-t009:** Principal component score, comprehensive score, and ranking of main flower visitors in *Camellia oleifera*.

Categories	Species	Pollen Load	Body Surface Characteristics	Body Length	Proportion	Visiting Frequency	PC1	PC2	F	Rank
Flies	*Stomorhina obsoleta*	−0.62	−0.84	−1.13	−0.37	−0.68	−0.93	0.47	−0.58	10
	*Lucilia sericata*	−0.66	−0.84	−0.59	−0.43	0.27	−0.58	−0.20	−0.49	8
	*Neomyia timorensis*	−0.64	−0.84	−0.43	−0.03	−0.59	−0.67	0.22	−0.45	6
	*Sarcophaga naemorrhoidalis*	−0.58	−0.84	−1.22	−0.46	−0.09	−0.81	0.30	−0.53	9
Hoverflies	*Phytomia zonata*	−0.09	0.70	0.58	−0.33	−0.06	0.22	−0.59	0.02	5
	*Episyrphus balteatus*	−0.70	−0.84	−0.49	−0.05	−1.64	−0.97	0.57	−0.59	11
	*Eristalinus arvorum*	−0.56	−0.84	−0.17	−0.40	−0.06	−0.53	−0.30	−0.47	7
	*Eristalis cerealis*	1.78	0.70	0.34	−0.23	−0.11	0.68	0.34	0.60	2
Bees	*Apis cerana*	0.02	1.47	0.24	−0.35	−0.50	0.27	−0.26	0.14	4
	*Apis mellifera*	2.11	1.47	0.47	2.98	1.48	2.13	1.79	2.04	1
Wasps	*Vespa velutina*	−0.07	0.70	2.39	−0.34	1.99	1.18	−2.33	0.31	3

**Table 10 insects-13-00539-t010:** Importance and weight of each variable of flower visitors.

Variables	Importance	Weight
Pollen load	5	0.3467
Body surface characteristics	4	0.2301
Body length	3	0.1460
Proportion	2	0.0870
Visiting frequency	1	0.0516

**Table 11 insects-13-00539-t011:** Normalised score, comprehensive score, and ranking of the main flower visitors to *Camellia oleifera*.

Categories	Species	Pollen Load	Body Surface Characteristics	Body Length	Proportion	Visiting Frequency	Total Score	Rank
Flies	*Stomorhina obsoleta*	0.03	0.00	0.03	0.02	0.27	0.03	11
	*Lucilia sericata*	0.02	0.00	0.17	0.01	0.53	0.07	8
	*Neomyia timorensis*	0.02	0.00	0.22	0.12	0.29	0.07	7
	*Sarcophaga naemorrhoidalis*	0.05	0.00	0.00	0.00	0.43	0.05	9
Hoverflies	*Phytomia zonata*	0.22	0.67	0.50	0.04	0.44	0.38	5
	*Episyrphus balteatus*	0.00	0.00	0.20	0.12	0.00	0.04	10
	*Eristalinus arvorum*	0.05	0.00	0.29	0.02	0.44	0.10	6
	*Eristalis cerealis*	0.88	0.67	0.43	0.07	0.42	0.65	2
Bees	*Apis cerana*	0.26	1.00	0.41	0.03	0.32	0.46	4
	*Apis mellifera*	1.00	1.00	0.47	1.00	0.86	0.91	1
Wasps	*Vespa velutina*	0.23	0.67	1.00	0.04	1.00	0.49	3

## Data Availability

The data that supports the findings of this study are available on request from the author.

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
