# Peer review of "What Are the Best Pollinator Candidates for Camellia oleifera: Do Not Forget Hoverflies and Flies"

_insects, 2022, doi:10.3390/insects13060539_

Round 1

Reviewer 1 Report

The authors offered to investigate the pollination biology of Camellia oleifera, questioning the diversity and efficiency of pollinator insects. This question is not yet solved for this economically important crop; even pollination is a main step of seeds and thus oil production. The authors used an interesting score matrix to evaluate insect efficiency, including insect behaviours, body parameters, and pollen load. Despite the interest in this question and the cleverness of the approaches, the authors did not evaluate the pollination efficiency of insects nor determine who was and who was not a pollinator. Still, they provided a very interesting preliminary study selecting the best pollinator candidates. They offered robust preparatory work for further research on C. oleifera pollinators and their efficiency. With at least some rewording, your paper will better reflect your work and will gain to be presented as preliminary results for further investigations. So, I have several remarks and questions; I hope I will give them in a way that will be constructive.

First of all, several points strongly reduce the power of the study and should be more clearly discussed : 

- you analysed pollen load from a pool of insects (washed/cleaned all-together in a single solution), so you reduced your replicate and lost inter-individual variability

- only one hour of insect observation per day, always at the same time of the day, so you missed all the possible interactions during the 23 other hours of a day. It could be interesting to develop more of your choice.

- my first more significant concern is the way you consider pollen load for pollination purposes. 1) you made a very interesting distinction between pollen and pseudo pollen grains on observed insects. It is not clear if you used the pollen value well to compare your insects, or maybe the ratio pollen/pseudo pollen (the pseudo pollen is not providing any pollination service) 2) Apis mellifera pollen load. All the pollen contained in the pollen pellets (pollen balls on the legs) is fully unavailable for pollination. It is no longer viable and physically glued in a ball. So this pollen must not be considered available pollen for pollination purposes. Even if these pollen pellets are easy to remove from honeybees before analysing insect pollen loads (available for pollination), how you managed your insects mixed everything. That created a huge artefact that explained the "leadership" position of Apis mellifera in your ordination.

- my second big concern is about "pollination efficiency" and "pollinators". Throughout your article, you used these words, but you never assessed pollination or pollination efficiency. A pollinating insect should not be able to collect as much pollen as possible but drop as much pollen as possible on the stigma (and pollen from the same species, pollen still viable, etc.). You didn't quantify at all the pollen deposition on stigma. You get closer to this point by recording insect behaviour, an essential and interesting record. Still, some of the observed insects are not pollinators. You can not present pollination efficiency without studying the pollination efficiency, so the link between insect visitation and seed production. 

=> Thus, your study, still very interesting, is not studying "pollination efficiency of wild pollinators of C. oleifera", but "What are the best pollinator candidates of Camelia oleifera: don't forget hoverflies and flies."

Please, find below my comments, line by line.

L.15. A. mellifère should be italics

L.17. the pollination OF flowering plants

L.23-26 Please provide full Latin names, as they were not previously presented.

L.22-23 this sentence has to be temperate… you give the impression that Apis mellifera had the higher pollen load available for pollination, but it is not the case as you included the pollen pellets

L.24-25 "the most efficient pollinator" you can not tell that as you didn't study the pollination efficiency in your study. Still, you can say that this species is the best pollinator candidate.

L.27 "wild A. melliferaI" how can you be sure they were wild? You provided not any explanation about that point, and you didn't explain how you could obtain such information. I do not know in China, but in Europe, we have no longer "wild" Apis mellifera. They can fly up to 10km for foraging. How could you be sure you had not a single honeybee hive in 10km around each of your studied fields? These honeybees were maybe not managed specifically for this crop, but they still can come from a surrounding apiary.

L.43 "declining number of pollinators" Declining population? Declining diversity?

L.69 Décembre 2021, it would be nice to discuss this absence of temporal replication in the discussion section

L.82 You conducted observation only from 11:00 to 12:00 am. Why such a choice? It is a very short period of observation. Why not try to observe all along the day to catch more insect diversity and behaviour diversity? And what about night flower visitation (by moths?), that also must be discussed well in the discussion section

L.88 "n=10" It is unclear how many specimens you measured. Ten only? 10 per species? Per site? Per group? I have the impression you should be more explicit.

L.93 "foraging behaviors" is a very important and interesting point and a real strength of your study. You should develop the behaviours recorded and how they are important to consider an insect as a good pollinator candidate.

L.90-101 a) Why didn't you use individual vials for insects? that would definitely give your data more power. That is also a point that you should explain/discuss

b) pollen load… why include everything in your study? For Apis mellifera, the pollen pellets are no longer available for pollination, so including this pollen in your pollen load observation seems to be a huge artefact. 

c) you have a quite specific plant species, with pollen and pseudo pollen. How did you take that into account? Only pollen can pollinate. The presence of pseudo pollen and the capacity of insects to carry pollen rather than pseudo pollen is a very important point in distinguishing good from bad pollinators candidates. This is also an important point to discuss in the discussion section and consider when considering the pollen load.

L.114 "to determine the pollinator efficiency" definitely not. You didn't determine any pollination efficiency as you didn't analyse nor the pollen deposition on stigmas nor the seed production linked to insect visitations. It is very important to review the way you present that point. From my point of view, giving that as determining the best pollinator candidate would be quite elegant.

 L.137 "early stage" What is a "stage"? The flower age? The beginning of the flowering period? The Neolithic? Please be more precise.

L.156 "often" and "most of" are vague… can you provide figures?

L.162 "grains per individual"… but that includes the pollen pellets… so this is not at all pollen grains available for pollination… This point should be clearly mentioned because it is a huge artefact in your analyses.

There also, it would be very interesting to report the ratio between pollen and pseudo pollen because here again, pseudo pollen will not pollinate at all.

L.121 "pollinators" still not. You only recorded them as flower visitors.

"prédominent"? Or would you mean "largest" regarding the body size? If yes, use larger and provide figures.

L.196 "pollination efficiency" still not tested…

L.197-198 "the score of A. mellifera was much higher than those of any other insect"… sure you included the pollen pellets… that is a HUGE artefact that should be strongly discussed.

L.218 "determine the efficient pollinator" you "aimed" maybe, but you didn't investigate that point

L.220 "pollinators" you don't know yet instead they are pollinators or not…

L.230 "the flower-pollinator relationship should be more finely evaluated" YES, definitely! And the pollinating role of each flower visitor is still to be established.

L.236 "most efficient pollinator" definitely not. You could say the "best pollinator candidate", but no more. But it is already a crucial stage, and your observations would definitely be the base for further work to determine the pollinator species of C. oleifera.

Figure 1. The text in white is not easy to read on white petals.

"pollinators", definitely not. They are "flower visitors", but you do not know if they are pollinators or not. 

What is the aim of pointing out the pollen baskets? How do you use this information?

Figure 2. It is surprising to represent "the body hairs of an insect" by such a punctual point as an arrow… the hairy surfaces are later than that… definitely and clearly, on Apis mellifera, there are several body areas hairier than the eye… 

"the remaining pollen", remaining after what? What is the interest of this info?

Table 1 in N°2, C. oleifera should be italics.

Table 2 "pollinators", definitely not, they are "flower visitors", but you do not know if they are pollinators or not. 

What is the total number of insects observed? Did you record individual insects or the number of flower visits? That is not clear and should be explained.

Table 3 "pollinators", definitely not, they are "flower visitors", but you do not know if they are pollinators or not. 

What is the total number of insects observed? Did you record individual insects or the number of flower visits? That is not clear and should be explained.

Latine names should be in italics.

Table 4 "pollinators", definitely not, they are "flower visitors", but you do not know if they are pollinators or not. 

"the main f" "f" should be removed, isn't it?

What is the number of observations (n) per species?

Table 5 "pollinators", definitely not, they are "flower visitors", but you do not know if they are pollinators or not. 

What is the number of observations (n) per species? I think that the ratio between pollen and pseudo pollen could be more informative than the sum of both and could give an idea of the chance to deposit pollen instead of pseudo pollen on the stigma.

Table 6 "flower -visiting insects " yes, that is definitely more realistic than pollinator!

What is n, the number of measured insects?

Table 8 Check alignment and capitals.

Table 9 "pollinators", definitely not, they are "flower visitors", but you do not know if they are pollinators or not. 

Table 11 "pollinators", definitely not, they are "flower visitors", but you do not know if they are pollinators or not. 

REFERENCES, please check that all Latin names are in italics

Author Response

Response to Reviewer 1 Comments

Dear reviewer,

On behalf of my co-authors, we thank you very much for giving us an opportunity to revise our manuscript (No. insects-1696754) again. We are very sorry for our incorrect in manuscript. We tried our best on revising the content of the manuscript and replying reviewer’s question. We apologize again for our negligence.

We have not considered submitting other journals. It will be an honor for all our authors to publish in “Insects”. We have revised our manuscript according reviewers’ comments which we hope to meet with approval. Special thanks to you for your patience and tolerance.

We will be happy to revise it again if our responses are not satisfied. Have a nice day.

Best regards!

Sincerely yours

Xiao-Ming Fan

On behalf of all authors.

------------------------------------------------------------------------------------------------------------------------------------

Point 1:You analysed pollen load from a pool of insects (washed/cleaned all-together in a single solution), so you reduced your replicate and lost inter-individual variability.

Response 1: Each species contains the same number of insects in each vial, each species has at least three vials, each vial is tested on three occasions. Three fields of view are randomly tested in each test, and the average value is recorded as a data.

Point 2: Only one hour of insect observation per day, always at the same time of the day, so you missed all the possible interactions during the 23 other hours of a day. It could be interesting to develop more of your choice.

Response 2: Nighttime is excluded because this article focuses on daytime pollination. 11:00 to 12:00 was selected because it was reported that the activity peak of Vespa velutina, Apis cerana and Phytomia zonata was from 11:00 to 12:00. At the same time, we made statistics on the visits of flies, aphid flies, bees and wasps per unit time (1 hour) in the oil-tea garden. The results are as follows. Therefore, we choose this period as the key period of research and investigation.

Figure 1 Visits of four kinds of flower-visiting insects in Camellia oleifera within one hour

Point 3: My first more significant concern is the way you consider pollen load for pollination purposes. 1) you made a very interesting distinction between pollen and pseudo pollen grains on observed insects. It is not clear if you used the pollen value well to compare your insects, or maybe the ratio pollen/pseudo pollen (the pseudo pollen is not providing any pollination service) 2) Apis mellifera pollen load. All the pollen contained in the pollen pellets (pollen balls on the legs) is fully unavailable for pollination. It is no longer viable and physically glued in a ball. So this pollen must not be considered available pollen for pollination purposes. Even if these pollen pellets are easy to remove from honeybees before analysing insect pollen loads (available for pollination), how you managed your insects mixed everything. That created a huge artefact that explained the "leadership" position of Apis mellifera in your ordination.

Response 3: 1) Based on your suggestion, the ratio of normal pollen to pseudopollen was added to the article and discussed in the discussion section. 2) I apologize for the misunderstanding caused by our fuzzy phrase in the article. In fact, once the bees were captured, we removed the pollen pellets from their pollen-carrying legs.The modifications based on your comments and the comments of the other two reviewers are as follows:

1):

3.2 Daily activity of insects visiting C. oleifera

There were many differences in the powder carrying capacity of the pollinator candidates (p < 0.05, Table 5). A. mellifera carried the largest amount, i.e. more than 50,000 grains per individual, followed by E. cerealis with 48,800 grains. Phytomia zonata, A. cerana, and V. velutina showed no significant difference. There were also significant differences in the proportion of normal and pseudopollen carried by different insects. The ratio of pseudopollen to normal pollen carried by most insects is about 0.1, and the ratio of flies is generally large, among which L. sericata is the highest, up to 0.43.

Table 5. Pollen-carrying situation of main flower visitors in Camellia oleifera.

Categories

Species

Pollen load (grain/individual)

Main powder-carrying position

Total amount of pollen

Normal pollen

Pseudopollen

Pseudopollen / Normal pollen

Flies

Stomorhina obsoleta

2320±129.84d

1700±169.97c

620±220.10c

0.38±0.16ab

Back plate and head

Lucilia sericata

1600±1074.97d

1200±1032.80c

400±699.21c

0.43±0.79a

Neomyia timorensis

2000±608.58d

1833±593.17c

167±175.68c

0.10±0.11cd

Scrcophaganae morrhoidalis

3188±400.64d

2688±512.62c

500±238.37c

0.21±0.16abcd

Hoverflies

Phytomia zonata

12667±237.17c

11333±210.82b

1500±241.96b

0.12±0.09cd

Body surface and feet

Episyrphus balteatus

733±262.94d

600±262.94c

133±172.13c

0.30±0.42abc

Back plate and head

Eristalis arvorum

3533±688.53d

3400±733.67c

133±172.13c

0.04±0.06d

Body surface

Eristalis cerealis

48800±5391.35b

48200±5202.56a

667±699.21c

0.01±0.02d

Villi on the body surface

Bees

Apis cerana

14650±1106.80c

13300±948.68b

1350±411.64b

0.10±0.03cd

Pollen-carrying legs and villi

Apis mellifera

55167±6549.81a

47583±6120.09a

7500±707.11a

0.16±0.02bcd

Wasps

Vespa velutina

13000±4216.37c

11675±4323.79b

1325±373.61b

0.13±0.05cd

Villi on the body surface

  1. Discussion

Additionally, we determined that E. cerealis is a ‘good’ candidate pollinator; P. zonata, A. cerana, and V. velutina are ‘ordinary’ candidate pollinators; and the remaining flower visitors were ‘inefficient’ candidate pollinators for C. oleifera. This division highlights the relative importance of diverse pollinators [38]. In this study, the most important indicator of the pollinator potential of flower visitors was the pollen load. However, unlike most plants, C. oleifera has pseudopollen, which is ineffective for pollination [41]. Therefore, we made a distinction between pollen and pseudopollen to better understand the effectiveness of the pollen carried by insects. The results showed that pseudopollen may not affect pollination by insects. Although insects carry pseudopollen during flower visits, the proportion of pseudopollen is approximately 0.1, which is lower than the 0.3 for pseudopollen to pollen on C. oleifera anthers [42]. This indicates that the collection of pseudopollen rather than pollen by insects may be selective; however, the criteria for this selection are not clear.

2):

2.6 Pollen load analysis

To quantify the pollen load capacity of pollinator candidates, we added a small amount of detergent to the visitor samples stored in the 10 mL vials to remove the pollen particles carried by them [20]. Then, we transferred 10 μL of the solution to a blood cell counting plate and counted, the tests were repeated at least ten times for each species. the numbers of normal pollen and pseudopollen separately using a microscope (OLYMPUS-BX51, Japan). It is worth noting that all the bees used for the test removed the pollen pellets beforehand. Pollen load was calculated as follows:

(Counted pollen number × 1000)/number of insects in each bottle

Point 4: My second big concern is about "pollination efficiency" and "pollinators". Throughout your article, you used these words, but you never assessed pollination or pollination efficiency. A pollinating insect should not be able to collect as much pollen as possible but drop as much pollen as possible on the stigma (and pollen from the same species, pollen still viable, etc.). You didn't quantify at all the pollen deposition on stigma. You get closer to this point by recording insect behaviour, an essential and interesting record. Still, some of the observed insects are not pollinators. You can not present pollination efficiency without studying the pollination efficiency, so the link between insect visitation and seed production.

Response 4: Thank you for your suggestion, we agree with that, and we've revised the inappropriate expression like "pollination efficiency" and "pollinator" in the manuscript.

Point 5: Thus, your study, still very interesting, is not studying "pollination efficiency of wild pollinators of C. oleifera", but "What are the best pollinator candidates of Camelia oleifera: don't forget hoverflies and flies."

Response 5: Thank you for your suggestion. The title of the section has been modified accordingly.

Point 6: L.15. A. mellifère should be italics

Response 6: Thank you for reporting our errors and we have addressed them accordingly. The specific modifications are as follows:

We found that Apis mellifera was the best candidate pollinator, but flies and hoverflies also played an important role in the pollination system.

Point 7: L.17. the pollination OF flowering plants

Response 7: Thank you for reporting our errors and we have addressed them accordingly.

Point 8: L.23-26 Please provide full Latin names, as they were not previously presented.

Response 8: Thank you for reporting our errors and we have addressed them based on your comments and the comments of the other two reviewers. The specific modifications are as follows:

Abstract: Camellia oleifera Abel is an important woody grain and oil plants, its pollination success is important for production. We conducted this study to select the best pollinator candidates of C. oleifera by principal component analysis and multi-attribute decision-making. Field observations of the flower-visiting characteristics of candidate pollinators were conducted at three sites. The insect species that visited flowers did not differ considerably between regions or time periods; however, the proportion of each species did vary. We recorded eleven main candidates from two orders and six families at the three sites. The pollen number carried on the of Apis mellifera was significantly higher than that of other insects; however, its visit frequency and body length were smaller than those of Vespa velutina. Statistical analysis shows that A. mellifera was the best candidate pollinator; Eristalis cerealis was a good candidate pollinator; Phytomia zonata, A. cerana, and V. velutina were ordinary candidate pollinators; and four fly species, Episyrphus balteatus, and E. arvorum were classified as inefficient candidate pollinators. Additionally, our study showed that flies and hoverflies also play an important role in the pollination system; hence, given the global decline in bee populations, they should also be considered in C. oleifera seed production.

Point 9: L.22-23 this sentence has to be temperate… you give the impression that Apis mellifera had the higher pollen load available for pollination, but it is not the case as you included the pollen pellets

Response 9: I am sorry that our unclear expression in the manuscript has caused you a misunderstanding. In fact, after the bees were caught, we removed the pollen pellets. Thank you for pointing out our mistakes and we have corrected them accordingly. The specific modifications are as follows:

The pollen number carried on the of Apis mellifera was significantly higher than that of other insects

Point 10: L.24-25 "the most efficient pollinator" you can not tell that as you didn't study the pollination efficiency in your study. Still, you can say that this species is the best pollinator candidate.

Response 10: I appreciate your pointing out our errors, and we've corrected them accordingly. Specific modifications include the following:

that A. mellifera was the best candidate pollinator; Eristalis cerealis was a good candidate pollinator; Phytomia zonata, A. cerana, and V. velutina were ordinary candidate pollinators; and four fly species, Episyrphus balteatus, and E. arvorum were classified as inefficient candidate pollinators.

Point 11:L. 27 "wild A. melliferaI" how can you be sure they were wild? You provided not any explanation about that point, and you didn't explain how you could obtain such information. I do not know in China, but in Europe, we have no longer "wild" Apis mellifera. They can fly up to 10km for foraging. How could you be sure you had not a single honeybee hive in 10km around each of your studied fields? These honeybees were maybe not managed specifically for this crop, but they still can come from a surrounding apiary.

Response 11: Thank you for pointing out the inappropriate exression, and we have revised them accordingly. The specific modifications are as follows:

Additionally, our study showed that flies and hoverflies also play an important role in the pollination system; hence, given the global decline in bee populations, they should also be considered in C. oleifera seed production.

Point 12: L.43 "declining number of pollinators" Declining population? Declining diversity?

Response 12: Our intention here is to express the decline in the number of insects, which we have modified accordingly. The specific modifications are as follows:

Understanding the pollination efficiency of pollinators can help us promote these interactions and predict the risk of plant extinction in the context of declining pollinator populations.

Point 13: L.69 Décembre 2021, it would be nice to discuss this absence of temporal replication in the discussion section.

Response 13: This period is selected because it is the blooming stage of C. oleifera.

Point 14: L.82 You conducted observation only from 11:00 to 12:00 am. Why such a choice? It is a very short period of observation. Why not try to observe all along the day to catch more insect diversity and behaviour diversity? And what about night flower visitation (by moths?), that also must be discussed well in the discussion section.

Response 14: Nighttime is excluded because this article focuses on daytime pollination. 11:00 to 12:00 is chosen because we have combined other articles and our previous pre-experiments and found that the number of insect visits in the park is the largest at this time. Obviously, we are in complete agreement with your suggestion. In future studies, we will further study the specific changes in insect visitation and behaviour at different periods. Thank you for asking this question, and to explain it, we have added a short explanation to the article:

We chose five sunny days in the first half of December at three sites and observed the flower-visiting insects from 11:00– 12:00 a.m., when visitors are most active

Figure 1 Visits of four kinds of flower-visiting insects in Camellia oleifera within one hour

Point 15: "n=10" It is unclear how many specimens you measured. Ten only? 10 per species? Per site? Per group? I have the impression you should be more explicit.

Response 15: This is to show that each species  has been measured ten times and the determination of the repetition amount refers to” Morphological traits as predictors of diet and microhabitat use in a diverse beetle assemblage”, ” Mayfly emergence production and body length response to hydrology in a tropical lowland stream” and “Pollination biology of Caragana sinica (Buchoz) Rehd”. We are sorry for the misexpression here and revised it in the manuscript:

we used vernier callipers to measure body length, head width and length, and back plate width and length of the collected visitor specimens, the measurements were repeated ten times for each species.

Point 16: L.93 "foraging behaviors" is a very important and interesting point and a real strength of your study. You should develop the behaviours recorded and how they are important to consider an insect as a good pollinator candidate.

Response 16: Thank you for your suggestion.

Similar to previous studies[15], we found that unlike bees and wasps, which are highly motivated to visit flowers and touch the stigma almost every time they visit flowers, flies are small in size, have a low pollen load and they are not very motivated to visit flowers, often perching on the anthers without contacting the stigma, which makes it difficult for them to pollinate the plants. However, they regulate the stability of the pollination system as generalised pollinators. In some plant species, flies also act as specialised pollinators [32,33]. Unlike flies, hoverflies have a high pollen load and most individuals will contact the stigma when they eat nectar, thereby transferring the pollen to the stigma to participate in pollination.  Hoverflies can increase the crop yield to the same extent as bees. Furthermore, hoverflies have certain advantages over bees, such as the ability to carry pollen further, thus promoting the movement of pollen and flow of genes throughout the landscape [34,35]. Therefore, attention should be paid to the pollination efficiency of flies and hoverflies considering the gradual decline in bee populations [36].

Point 17: L.90-101 a) Why didn't you use individual vials for insects? that would definitely give your data more power. That is also a point that you should explain/discuss. b) pollen load… why include everything in your study? For Apis mellifera, the pollen pellets are no longer available for pollination, so including this pollen in your pollen load observation seems to be a huge artefact. c) you have a quite specific plant species, with pollen and pseudo pollen. How did you take that into account? Only pollen can pollinate. The presence of pseudo pollen and the capacity of insects to carry pollen rather than pseudo pollen is a very important point in distinguishing good from bad pollinators candidates. This is also an important point to discuss in the discussion section and consider when considering the pollen load.

Response 17:

  1. a) Each species contains the same number of insects in each vial, each species has at least three vials, each vial is tested on three occasions. Three fields of view are randomly tested in each test, and the average value is recorded as a data. We will try to overcome this proble in future experiments, and thank you for your suggestion again.
  2. b) I'm sorry, this is our inappropriate expression in the section. We consciously removed the pollen pellets carried by the bees after catching the bees, but we have not expressed this clearly in the article. I'm awfully sorry.
  3. c) Thank you for your suggestion. We have provided a supplementary explanation in the discussion on this issue. Pseudopollen is indee the focus of our research group at present, and we will strive to obtain further research results in the next work.
  4. Discussion

Additionally, we determined that E. cerealis is a ‘good’ candidate pollinator; P. zonata, A. cerana, and V. velutina are ‘ordinary’ candidate pollinators; and the remaining flower visitors were ‘inefficient’ candidate pollinators for C. oleifera. This division highlights the relative importance of diverse pollinators [38]. In this study, the most important indicator of the pollinator potential of flower visitors was the pollen load. However, unlike most plants, C. oleifera has pseudopollen, which is ineffective for pollination [41]. Therefore, we made a distinction between pollen and pseudopollen to better understand the effectiveness of the pollen carried by insects. The results showed that pseudopollen may not affect pollination by insects. Although insects carry pseudopollen during flower visits, the proportion of pseudopollen is approximately 0.1, which is lower than the 0.3 for pseudopollen to pollen on C. oleifera anthers [42]. This indicates that the collection of pseudopollen rather than pollen by insects may be selective; however, the criteria for this selection are not clear.

Point 18: L.114 "to determine the pollinator efficiency" definitely not. You didn't determine any pollination efficiency as you didn't analyse nor the pollen deposition on stigmas nor the seed production linked to insect visitations. It is very important to review the way you present that point. From my point of view, giving that as determining the best pollinator candidate would be quite elegant.

Response 18: Thank you for pointing out our mistakes and we have corrected them accordingly. The specific modifications are as follows:

2.7.3 Principal component analysis

To determine the best pollinator candidate, the comprehensive score of each insect was obtained by principal component analysis (PCA). The data were standardised to a mean of zero and variance of one before conducting the PCA. The contribution of each feature to different principal components. According to Gutten’s lower bound principle, eigenvalues <1 were ignored [22].

Point 19: L.137 "early stage" What is a "stage"? The flower age? The beginning of the flowering period? The Neolithic? Please be more precise.

Response 19: The “early stage” here refers to the first half of December, and we have made corresponding changes in the manuscipt. The following specific amendments are made.

In the first half of December, wasps were the main insects, but in the second half of December, the proportions of wasps and bees fluctuated greatly. Flies and hoverflies were found in a large proportion (Table 3).

Point 20: L.156 "often" and "most of" are vague… can you provide figures?

Response 20: Thank you for suggesting this. Due to the relatively large number and species of insects, we failed to find a qunatitative method which can provide figures. We will pay attention to this point in future research. Nonetheless, we have made modifications to be more accurate. Here are the changes:

There were significant differences in the behaviours of different insects. Flies spend more than half of their time visiting anthers and inhabited the anther instead of the stigma. Among the hoverflies, most showed foraging behaviours and frequently made contact with the stigmas (more than half of the hoverflies will come into contact with the stigma in the process of observation), except Episyrphus balteatus who often hovered over but did not visit the flowers. Most of the bees visited for collecting nectar, but A. mellifera showed higher enthusiasm and actively collected pollen in their pollen-carrying legs. The wasps were very active, usually buried their heads in the anthers to forage, and contact with the stigmas almost every time (Table 4).

Table 4. The main foraging behaviour of the main flower visitors of Camellia oleifera.

Categories

Species

Time for each visiting flower (s)

Single flower visit frequency (times/min)

Main foraging behaviour

The shortest

The longest

The lowest

The highest

Average frequency

Flies

Stomorhina obsoleta

9

96

1

5

2.17±1.24c

Spending more than half of their visiting anthers more, but touch the stigma less.

Lucilia sericata

16

22

2

4

3.00±0.77abc

Neomyia timorensis

8

160

1

5

2.25±1.16c

Scrcophaganae morrhoidalis

8

54

1

6

2.69±1.38bc

Hoverflies

Phytomia zonata

3

270

1

5

2.71±1.28bc

More active and more than half of them will touch the stigma.

Episyrphus balteatus

7

21

1

2

1.33±0.47c

Mainly inspect the flowers and stay for a short time.

Eristalis arvorum

5

78

1

6

2.71±1.28bc

Sometimes will rest on flowers temporarily without any activities.

Eristalis cerealis

5

34

1

3

2.67±0.60bc

Taking a short time to forage, but they will touch the stigma almost every time.

Bees

Apis cerana

3

46

1

3

2.33±0.58bc

Visiting time is short, and the enthusiasm of visiting flowers is low.

Apis mellifera

2

148

1

7

4.06±1.61ab

Visiting flower frequently, collecting pollen actively, and contacting stigma almost every time.

Wasps

Vespa velutina

2

127

1

6

4.47±1.27a

Visiting flower actively, and touch stigma almost every time.

Within columns, different letters indicate significant differences at p < 0.05.

Point 21: L.162 "grains per individual"… but that includes the pollen pellets… so this is not at all pollen grains available for pollination… This point should be clearly mentioned because it is a huge artefact in your analyses. There also, it would be very interesting to report the ratio between pollen and pseudo pollen because here again, pseudo pollen will not pollinate at all.

Response 21: Thank for your continued attention to this point again. After the bees were caught, we took out the pollen pellets, and we're sorry again for our inaccurate expression. Based on your suggestion, we added the radio of normal pollen to pseudopollen in the article and talked about it in the discussion section. The modifications are as follows:

Table 5. Pollen-carrying situation of main flower visitors in Camellia oleifera.

Categories

Species

Pollen load (grain/individual)

Main powder-carrying position

Total amount of pollen

Normal pollen

Pseudopollen

Pseudopollen / Normal pollen

Flies

Stomorhina obsoleta

2320±129.84d

1700±169.97c

620±220.10c

0.38±0.16ab

Back plate and head

Lucilia sericata

1600±1074.97d

1200±1032.80c

400±699.21c

0.43±0.79a

Neomyia timorensis

2000±608.58d

1833±593.17c

167±175.68c

0.10±0.11cd

Scrcophaganae morrhoidalis

3188±400.64d

2688±512.62c

500±238.37c

0.21±0.16abcd

Hoverflies

Phytomia zonata

12667±237.17c

11333±210.82b

1500±241.96b

0.12±0.09cd

Body surface and feet

Episyrphus balteatus

733±262.94d

600±262.94c

133±172.13c

0.30±0.42abc

Back plate and head

Eristalis arvorum

3533±688.53d

3400±733.67c

133±172.13c

0.04±0.06d

Body surface

Eristalis cerealis

48800±5391.35b

48200±5202.56a

667±699.21c

0.01±0.02d

Villi on the body surface

Bees

Apis cerana

14650±1106.80c

13300±948.68b

1350±411.64b

0.10±0.03cd

Pollen-carrying legs and villi

Apis mellifera

55167±6549.81a

47583±6120.09a

7500±707.11a

0.16±0.02bcd

Wasps

Vespa velutina

13000±4216.37c

11675±4323.79b

1325±373.61b

0.13±0.05cd

Villi on the body surface

  1. Discussion

Additionally, we determined that E. cerealis is a ‘good’ candidate pollinator; P. zonata, A. cerana, and V. velutina are ‘ordinary’ candidate pollinators; and the remaining flower visitors were ‘inefficient’ candidate pollinators for C. oleifera. This division highlights the relative importance of diverse pollinators [38]. In this study, the most important indicator of the pollinator potential of flower visitors was the pollen load. However, unlike most plants, C. oleifera has pseudopollen, which is ineffective for pollination [41]. Therefore, we made a distinction between pollen and pseudopollen to better understand the effectiveness of the pollen carried by insects. The results showed that pseudopollen may not affect pollination by insects. Although insects carry pseudopollen during flower visits, the proportion of pseudopollen is approximately 0.1, which is lower than the 0.3 for pseudopollen to pollen on C. oleifera anthers [42]. This indicates that the collection of pseudopollen rather than pollen by insects may be selective; however, the criteria for this selection are not clear.

Point 22: L.121 "pollinators" still not. You only recorded them as flower visitors. "prédominent"? Or would you mean "largest" regarding the body size? If yes, use larger and provide figures.

Response 22: Thank you for pointing out our mistakes and we have corrected them accordingly. The specific modifications are as follows:

3.3 Body measurements of main visiting insects

Among the 11 main pollinator candidates, V. velutina was the largest species, its body length reached 20.42 mm. Bees and hoverflies are very similar in appearance, but the bees were larger in size (p < 0.05, Table 6).

Point 23: L.196 "pollination efficiency" still not tested…

Response 23: Thank you for pointing out our mistakes and we have corrected them accordingly. The specific modifications are as follows:

3.4 Determination of the best pollinator candidate

Using correlation analysis, we selected five factors with high correlation from the six previously recorded factors for follow-up analysis (p < 0.01, Table 7). PCA with eigenvalues >1 contributed to 82.66% of the total cumulative variance (Table 8). Then, we calculated the comprehensive scores of 11 species; only the top five insects had positive scores (Table 9). In the multi-attribute decision-making (MADM) model, the values for the decision matrix and weight of five variables of the dominant flower visitors were calculated by considering the influence of each factor on pollinator candidates (Table 10). Thus, we obtained the scores for each insect species; the score of A. mellifera was much higher than those of other insects, reaching 0.91 (Table 11).

Point 24: L.197-198 "the score of A. mellifera was much higher than those of any other insect"… sure you included the pollen pellets… that is a HUGE artefact that should be strongly discussed.

Response 24: Thank you for your reminder again. We did take this into account in our experiments. I think we forgot to mention that after the bees were caught, we took out the pollen pellets. In fact, if the pollen pellets were not removed, the pollen load data would be much larger.

Point 25: L.218 "determine the efficient pollinator" you "aimed" maybe, but you didn't investigate that point

Response 25: Thank you for pointing out our mistakes and combining your views with the another reviewer, we chose to delete the paragraph where the sentence is located.

Point 26: L.220 "pollinators" you don't know yet instead they are pollinators or not…

Response 26: Thank you for pointing out our mistakes and we have corrected them accordingly. The specific modifications are as follows:

This suggested that the classification of flower visitors into different functional groups should take into consideration the contribution of (different) flower visitors to pollination, while disregarding taxonomic affinities [25,26].

Point 27: L.230 "the flower-pollinator relationship should be more finely evaluated" YES, definitely! And the pollinating role of each flower visitor is still to be established.

Response 27: Thanks for pointing this out, we will investigate this further in future experiments.

Point 28: L.236 "most efficient pollinator" definitely not. You could say the "best pollinator candidate", but no more. But it is already a crucial stage, and your observations would definitely be the base for further work to determine the pollinator species of C. oleifera.

Response 28: Thank you for pointing out our mistakes and we have corrected them accordingly. The specific modifications are as follows:

We used statistical methods to select A. mellifera as the best C. oleifera candidate pollinator, and found that flies and hoverflies are also play a very important role in the pollination system.

Additionally, we determined that E. cerealis is a ‘good’ candidate pollinator; P. zonata, A. cerana, and V. velutina are ‘ordinary’ candidate pollinators; and the remaining flower visitors were ‘inefficient’ candidate pollinators for C. oleifera. This division highlights the relative importance of diverse pollinators [38].

Point 29: Figure 1. The text in white is not easy to read on white petals. "Pollinators", definitely not. They are "flower visitors", but you do not know if they are pollinators or not. What is the aim of pointing out the pollen baskets? How do you use this information?

Response 29: I appreciate your suggestion. Based on your suggestion, we have adjusted the figure 1 accordingly. We really have not fugure out what the aim of the pollen baskets of A. mellifera. Than you for provied a good idea for us to investigate further.

Figure 1. Flower visitors of Camellia oleifera.

Point 30: Figure 2. It is surprising to represent "the body hairs of an insect" by such a punctual point as an arrow… the hairy surfaces are later than that… definitely and clearly, on Apis mellifera, there are several body areas hairier than the eye…  "the remaining pollen", remaining after what? What is the interest of this info?

Response 30: The original intent of using arrows is for readers to better identify body hair, but we are very sorry that our method of using arrows to indicate the location of body hair didn't work as well. The remaining pollen refers to the residual pollen on the surface of the body after removal from the bottle. We think that this can better show the location of the pollen carried by insects, so we regret that we did not take pictures of all the pollen left on the surface of insects. We apologise again for this.

Figure 2. Posture of Camellia oleifera flower visitors. The red arrow represents the remaining pollen on the surface of the insect after taking it out of the penicillin bottle. Bar=1mm.

Point 31: Table 1 in N°2, C. oleifera should be italics

Response 31: Thank you for pointing out our mistakes and we have corrected them accordingly. The specific modifications are as follows:

Table 1. Distribution of Camellia oleifera in the study sites in Changsha, China and sampling methods employed for different sites.

No.

Location

Latitude and longitude

Basic situation

Sampling methods

1

Tianxin District

28°8′14″ N,

112°59′08″ E

Only small trees, many varieties, and well managed.

Trees in east, west, south, north, and middle of the areas were selected.

2

Yuelu District

28°13′48″ N, 112°55′53″ E

Neat arrangement of trees but low planting density. Predominance of young C. oleifera trees.

One row each in the upper, middle, and lower parts of the terrace was selected; trees in the left, right, and centre of each row were targeted.

3

Wangcheng District

28°22′12″ N, 112°49′12″ E

Covers a large area; located far away from urban areas; many tall trees.

Eight plots of 10 m × 10 m were set up; trees in the east, west, south, north, and middle were selected.

Point 32: Table 2 "pollinators", definitely not, they are "flower visitors", but you do not know if they are pollinators or not. What is the total number of insects observed? Did you record individual insects or the number of flower visits? That is not clear and should be explained.

Response 32: Thank you for pointing out our mistakes and we have corrected them based on your comments and the comments of the other two reviewers. The specific modifications are as follows:

2.3 Insect proportion

To investigate the flower visitors visits to C. oleifera during the flowering period (from November to January), we conducted the following experiments in December (the temperature in the first half of the month was 6–16 ℃ and in the second half was 3–11 ℃). We observed 2679 insects in the process. (a) We chose five sunny days in the first half of December at three sites and observed the flower-visiting insects from 11:00 a.m.–12:00pm., when visitors are most active [18]; (b) we observed the visits to each site in the second half of December. The proportions of visiting insect species were calculated using the following formula:

Insect proportion = Single species insect number/total insect number

Table 2. Abundance of flower visitors of Camellia oleifera of three sites in first half of December.

NO.

Order

Family

Species

Proportion

Site1

Site2

Site3

A

Diptera

Calliphoridae

Stomorhina obsoleta

12.14%

9.74%

0.19%

B

Diptera

Calliphoridae

Lucilia sericata

5.00%

4.62%

1.81%

C

Diptera

Muscidae

Neomyia timorensis

4.29%

12.31%

8.53%

D

Diptera

Sarcophagidae

Scrcophaganae morrhoidalis

11.43%

4.10%

0.56%

E

Diptera

Syrphidae

Phytomia zonata

11.43%

8.72%

1.72%

F

Diptera

Syrphidae

Episyrphus balteatus

1.43%

6.67%

8.72%

G

Diptera

Syrphidae

Eristalis arvorum

8.57%

9.74%

1.39%

H

Diptera

Syrphidae

Eristalis cerealis

2.86%

3.08%

5.98%

I

Hymenoptera

Apidae

Apis cerana

2.14%

5.13%

3.20%

J

Hymenoptera

Apidae

Apis mellifera

3.57%

33.33%

66.06%

K

Hymenoptera

Vespidae

Vespa mandarinia

0.00%

1.03%

0.46%

L

Hymenoptera

Vespidae

Vespa velutina

37.14%

1.54%

1.39%

Point 33: Table 3 "pollinators", definitely not, they are "flower visitors", but you do not know if they are pollinators or not. What is the total number of insects observed? Did you record individual insects or the number of flower visits? That is not clear and should be explained. Latine names should be in italics.

Response 33: Thank you for pointing out our mistakes and we have corrected them based on your comments and the comments of the other two reviewers. The specific modifications are as follows:

2.3 Insect proportion

To investigate the flower visitors visits to C. oleifera during the flowering period (from November to January), we conducted the following experiments in December (the temperature in the first half of the month was 6–16 ℃ and in the second half was 3–11 ℃). We observed 2679 insects in the process. (a) We chose five sunny days in the first half of December at three sites and observed the flower-visiting insects from 11:00 a.m.–12:00pm., when visitors are most active [18]; (b) we observed the visits to each site in the second half of December. The proportions of visiting insect species were calculated using the following formula:

Insect proportion = Single species insect number/total insect number

Table 3. The proportion of Camellia oleifera flower visitors in different periods in site 1.

Categories

Family

Species

Proportion

First half of December

Second half of December

Volatility

Flies

Calliphoridae

Stomorhina obsoleta

12.14%

20.60%

8.46%

Lucilia sericata

5.00%

1.01%

-3.99%

Muscidae

Neomyia timorensis

4.29%

8.04%

3.75%

Sarcophagidae

Scrcophaganae morrhoidalis

11.43%

4.52%

-6.91%

Hoverflies

Syrphidae

Phytomia zonata

11.43%

15.08%

3.65%

Episyrphus balteatus

1.43%

10.05%

8.62%

Eristalis arvorum

8.57%

4.52%

-4.05%

Eristalis cerealis

2.86%

3.02%

0.16%

Bees

Apidae

Apis cerana

2.14%

4.02%

1.88%

Apis mellifera

3.57%

22.61%

19.04%

Wasps

Vespidae

Vespa velutina

37.14%

6.53%

-30.61%

Point 34: Table 4 "pollinators", definitely not, they are "flower visitors", but you do not know if they are pollinators or not. "The main f" "f" should be removed, isn't it? What is the number of observations (n) per species?

Response 34: Thank you for pointing out our mistakes and we have corrected them based on your comments and the comments of the other two reviewers. The specific modifications are as follows:

2.5 Foraging behaviours

To explore the behavioural differences among the pollinator candidates, we made the following records: (a) visiting time, i.e. duration from the insect landing on the flower to the insect leaving, we recorded 205 insects in the process; (b) visiting frequency, i.e. number of insect visits to the flowers in one min, we recorded 168 insects in the process; and (c) foraging behaviours [19].

Table 4. The main foraging behaviour of the main flower visitors of Camellia oleifera.

Categories

Species

Time for each visiting flower (s)

Single flower visit frequency (times/min)

Main foraging behaviour

The shortest

The longest

The lowest

The highest

Average frequency

Flies

Stomorhina obsoleta

9

96

1

5

2.17±1.24c

Spending more than half of their visiting anthers more, but touch the stigma less.

Lucilia sericata

16

22

2

4

3.00±0.77abc

Neomyia timorensis

8

160

1

5

2.25±1.16c

Scrcophaganae morrhoidalis

8

54

1

6

2.69±1.38bc

Hoverflies

Phytomia zonata

3

270

1

5

2.71±1.28bc

More active and more than half of them will touch the stigma.

Episyrphus balteatus

7

21

1

2

1.33±0.47c

Mainly inspect the flowers and stay for a short time.

Eristalis arvorum

5

78

1

6

2.71±1.28bc

Sometimes will rest on flowers temporarily without any activities.

Eristalis cerealis

5

34

1

3

2.67±0.60bc

Taking a short time to forage, but they will touch the stigma almost every time.

Bees

Apis cerana

3

46

1

3

2.33±0.58bc

Visiting time is short, and the enthusiasm of visiting flowers is low.

Apis mellifera

2

148

1

7

4.06±1.61ab

Visiting flower frequently, collecting pollen actively, and contacting stigma almost every time.

Wasps

Vespa velutina

2

127

1

6

4.47±1.27a

Visiting flower actively, and touch stigma almost every time.

Within columns, different letters indicate significant differences at p < 0.05.

Point 35: Table 5 "pollinators", definitely not, they are "flower visitors", but you do not know if they are pollinators or not. What is the number of observations (n) per species? I think that the ratio between pollen and pseudo pollen could be more informative than the sum of both and could give an idea of the chance to deposit pollen instead of pseudo pollen on the stigma

Response 35: Thank you for pointing out our mistakes and we have corrected them based on your comments and the comments of the other two reviewers. The specific modifications are as follows:

2.6 Pollen load analysis

To quantify the pollen load capacity of pollinator candidates, we added a small amount of detergent to the visitor samples stored in the 10 mL vials to remove the pollen particles carried by them [20]. Then, we transferred 10 μL of the solution to a blood cell counting plate and counted, the tests were repeated at least ten times for each species. the numbers of normal pollen and pseudopollen separately using a microscope (OLYMPUS-BX51, Japan). It is worth noting that all the bees used for the test removed the pollen pellets beforehand. Pollen load was calculated as follows:

(Counted pollen number × 1000)/number of insects in each bottle

Table 5. Pollen-carrying situation of main flower visitors in Camellia oleifera.

Categories

Species

Pollen load (grain/individual)

Main powder-carrying position

Total amount of pollen

Normal pollen

Pseudopollen

Pseudopollen / Normal pollen

Flies

Stomorhina obsoleta

2320±129.84d

1700±169.97c

620±220.10c

0.38±0.16ab

Back plate and head

Lucilia sericata

1600±1074.97d

1200±1032.80c

400±699.21c

0.43±0.79a

Neomyia timorensis

2000±608.58d

1833±593.17c

167±175.68c

0.10±0.11cd

Scrcophaganae morrhoidalis

3188±400.64d

2688±512.62c

500±238.37c

0.21±0.16abcd

Hoverflies

Phytomia zonata

12667±237.17c

11333±210.82b

1500±241.96b

0.12±0.09cd

Body surface and feet

Episyrphus balteatus

733±262.94d

600±262.94c

133±172.13c

0.30±0.42abc

Back plate and head

Eristalis arvorum

3533±688.53d

3400±733.67c

133±172.13c

0.04±0.06d

Body surface

Eristalis cerealis

48800±5391.35b

48200±5202.56a

667±699.21c

0.01±0.02d

Villi on the body surface

Bees

Apis cerana

14650±1106.80c

13300±948.68b

1350±411.64b

0.10±0.03cd

Pollen-carrying legs and villi

Apis mellifera

55167±6549.81a

47583±6120.09a

7500±707.11a

0.16±0.02bcd

Wasps

Vespa velutina

13000±4216.37c

11675±4323.79b

1325±373.61b

0.13±0.05cd

Villi on the body surface

Within columns, different letters indicate significant differences at p < 0.05.

Point 36: Table 6 "flower -visiting insects " yes, that is definitely more realistic than pollinator! What is n, the number of measured insects?

Response 36: Thank you for pointing out our mistakes and we have corrected them based on your comments and the comments of the other two reviewers. On the subject of n we have continued the concrete explanation in the material and method. The specific modifications are as follows:

2.4 Body measurements of main pollinator candidates

For comparative analysis of the flower visitors’ body characteristics, we used vernier callipers to measure body length, head width and length, and back plate width and length of the collected visitor specimens, the measurements were repeated ten times for each species. In addition, we photographed the main flower visitors using a stereo microscope (OLYMPUS-SZX16, Japan).

Table 6. Posture characteristics of main flower-visiting insects in Camellia oleifera.

Categories

Species

Body length (mm)

Head width (mm)

Head length (mm)

Shoulder length (mm)

Shoulder width (mm)

Body surface characteristics

Flies

Stomorhina obsoleta

6.95±1.17f

2.28±0.51fg

2.01±0.43ef

2.47±0.58f

2.17±0.56h

Body surface bristles

Lucilia sericata

8.99±0.88e

3.44±0.33e

2.18±0.24def

3.62±0.37de

3.44±0.42cd

Neomyia timorensis

9.63±0.95de

3.58±0.24de

2.46±0.60cde

3.65±0.41de

3.25±0.26de

Scrcophaganae morrhoidalis

6.60±0.45f

2.19±0.35g

1.85±0.42f

2.66±0.21f

2.77±0.36fg

Bristled, sparse at the back

Hoverflies

Phytomia zonata

13.49±0.60b

4.99±0.18b

2.81±0.31c

4.82±0.41b

5.07±0.36b

Densely tomentose

Episyrphus balteatus

9.39±0.63e

2.57±0.19f

2.68±0.41cd

2.61±0.14f

2.43±0.10gh

Tomentose on both sides and short hairs on the ventral segment

Eristalis arvorum

10.63±1.06d

3.88±0.14cd

3.57±0.12b

3.98±0.27cd

3.77±0.38c

Dorsal plate is tomentose and the ventral segment is short-haired

Eristalis cerealis

12.58±1.65bc

4.15±0.91c

2.76±0.32c

4.12±0.56c

3.59±1.03cd

Densely tomentose, and the dorsal plate is particularly dense

Bees

Apis cerana

12.20±0.73c

3.66±0.18de

2.91±0.72c

3.70±0.31de

3.04±0.33ef

Densely covered with yellow villi, short ventral hairs, with pollen-carrying legs

Apis mellifera

13.05±0.65bc

3.45±0.24e

2.56±0.71cd

3.54±0.34e

2.97±0.57ef

Wasps

Vespa velutina

20.42±2.48a

5.34±0.26a

4.33±1.01a

6.06±0.47a

6.67±0.51a

Densely tomentose

Within columns, different letters indicate significant differences at p < 0.05.

Point 37: Table 8 Check alignment and capitals.

Response 37: Thank you for reporting our errors and we have addressed them accordingly. The specific modifications are as follows:

Table 8. Score coefficient, variance contribution and cumulative contribution rate of the two principal components.

Principal component

Pollen load

Body surface characteristics

Body length

Proportion

Visiting frequency

Eigenvalue

Variance contribution

Cumulative contribution rates

PC1

0.27

0.28

0.24

0.22

0.25

3.11

62.19%

62.19%

PC2

0.38

-0.04

-0.60

0.61

-0.32

1.02

20.47%

82.66%

Point 38: Table 9 "pollinators", definitely not, they are "flower visitors", but you do not know if they are pollinators or not.

Response 38: Thank you for pointing out our mistakes, and we have modified what you mentioned based on your comments and the comments of the other two reviewers. The specific modifications are as follows:

Table 9. Principal component score, comprehensive score and ranking of main flower visitors in Camellia oleifera.

Categories

Species

Pollen load

Body surface characteristics

Body length

Proportion

Visiting frequency

PC1

PC2

F

Rank

Flies

Stomorhina obsoleta

-0.62

-0.84

-1.13

-0.37

-0.68

-0.93

0.47

-0.58

10

Lucilia sericata

-0.66

-0.84

-0.59

-0.43

0.27

-0.58

-0.20

-0.49

8

Neomyia timorensis

-0.64

-0.84

-0.43

-0.03

-0.59

-0.67

0.22

-0.45

6

Scrcophaganae morrhoidalis

-0.58

-0.84

-1.22

-0.46

-0.09

-0.81

0.30

-0.53

9

Hoverflies

Phytomia zonata

-0.09

0.70

0.58

-0.33

-0.06

0.22

-0.59

0.02

5

Episyrphus balteatus

-0.70

-0.84

-0.49

-0.05

-1.64

-0.97

0.57

-0.59

11

Eristalis arvorum

-0.56

-0.84

-0.17

-0.40

-0.06

-0.53

-0.30

-0.47

7

Eristalis cerealis

1.78

0.70

0.34

-0.23

-0.11

0.68

0.34

0.60

2

Bees

Apis cerana

0.02

1.47

0.24

-0.35

-0.50

0.27

-0.26

0.14

4

Apis mellifera

2.11

1.47

0.47

2.98

1.48

2.13

1.79

2.04

1

Wasps

Vespa velutina

-0.07

0.70

2.39

-0.34

1.99

1.18

-2.33

0.31

3

Point 39: Table 11 "pollinators", definitely not, they are "flower visitors", but you do not know if they are pollinators or not.

Response 39: Thank you for reporting our errors and we have addressed them based on your comments and the comments of the other two reviewers. The specific modifications are as follows:

Table 11. Normalised score, comprehensive score and ranking of main flower visitors of Camellia oleifera.

Categories

Species

Pollen load

Body surface characteristics

Body length

Proportion

Visiting frequency

Total score

Rank

Flies

Stomorhina obsoleta

0.03

0.00

0.03

0.02

0.27

0.03

11

Lucilia sericata

0.02

0.00

0.17

0.01

0.53

0.07

8

Neomyia timorensis

0.02

0.00

0.22

0.12

0.29

0.07

7

Scrcophaganae morrhoidalis

0.05

0.00

0.00

0.00

0.43

0.05

9

Hoverflies

Phytomia zonata

0.22

0.67

0.50

0.04

0.44

0.38

5

Episyrphus balteatus

0.00

0.00

0.20

0.12

0.00

0.04

10

Eristalis arvorum

0.05

0.00

0.29

0.02

0.44

0.10

6

Eristalis cerealis

0.88

0.67

0.43

0.07

0.42

0.65

2

Bees

Apis cerana

0.26

1.00

0.41

0.03

0.32

0.46

4

Apis mellifera

1.00

1.00

0.47

1.00

0.86

0.91

1

Wasps

Vespa velutina

0.23

0.67

1.00

0.04

1.00

0.49

3

Point 40: REFERENCES, please check that all Latin names are in italics

Response 40: Thank you for pointing out our mistakes and we have corrected them accordingly in the manuscript. The specific modifications are as follows:

  1. Li, Y.; Liu, K.; Zhu, J.; Jiang, Y.; Huang, Y.; Zhou, Z.; Chen, C.; Yu, F. Manganese accumulation and plant physiology behavior of Camellia oleifera in response to different levels of nitrogen fertilization. Ecotoxicology and Environmental Safety 2019, 184, 109603, doi:10.1016/j.ecoenv.2019.109603.
  2. Xiong, H.; Zou, F.; Yuan, D.; Tan, X.; Niu, G. Comparison of self- and cross-pollination in pollen tube growth, early ovule development and fruit set of Camellia grijsii. International Journal of Agriculture and Biology 2019, 21, 819-826, doi:10.17957/IJAB/15.0960.
  3. Chao, G.; Yuan, D.; Yang, Y.; Wang, B. Pollen Tube Growth and Double Fertilization in Camellia oleifera. Journal of the American Society for Horticultural Science American Society for Horticultural Science 2015, 140, 12-18, doi:10.1252/kakoronbunshu.29.576.
  4. Li, H.-Y.; Luo, A.-C.; Hao, Y.-J.; Dou, F.-Y.; Kou, R.-M.; Orr, M.C.; Zhu, C.-D.; Huang, D.-Y. Comparison of the pollination efficiency of Apis cerana with wild bees in oil-seed camellia fields. Basic and Applied Ecology 2021, 56, 250-258, doi:10.1016/j.baae.2021.08.005.
  5. Su, R.; Dong, Y.; Dong, K.; He, S. The toxic honey plant Camellia oleifera. Journal of Apicultural Research 2012, 51, 277-279, doi:10.3896/IBRA.1.51.3.09.
  6. Huang, D.Y.; Hao, J.S.; Yu, J.F.; Zhang, Y.Z.; Zhu, C.D. Review on Camellia oleifera. Life Science Research 2009, 13, 459-465, doi:10.16605/j.cnki.1007-7847.2009.05.012.
  7. Wu, T.; Tang, J.; Huang, S.Q. Foraging behavior and pollination efficiency of generalist insects in an understory dioecious shrub Helwingia japonica. American Journal of Botany 2020, 107, doi:10.1002/ajb2.1524.
  8. Wei, W.; Wu, H.; Li, X.; Wei, X.; Lu, W.; Zheng, X. Diversity, Daily Activity Patterns, and Pollination Effectiveness of the Insects Visiting Camellia osmantha, C. vietnamensis, and C. oleifera in South China. Insects 2019, 10, doi:10.3390/insects10040098.
  9. Seid, E.; Mohammed, W.; Abebe, T. Genetic Diversity Assessment through Cluster and Principal Component Analysis in Potato (Solanum tuberosum L.) Genotypes for Processing Traits. International Journal of Food Science and Agriculture 2021, 5, doi:10.26855/ijfsa.2021.09.014.
  10. Ji, T.; Wei, M.; Li, Z. Some ideas to solve the problem of Camellia oleifera pollination. South China Forestry Science 2018, 46, 22-25+34, doi:10.16259/j.cnki.36-1342/s.2018.03.007.

Reviewer 2 Report

  Pollination Efficiency of wild pollinators of Camellia oleifera:  don’t forget hoverflies and flies

  1. Title needs to be changed to be more meaningful .
  2. Importance of  f Camellia oleifera plant shall be added as a line in abstract - as scope or background of research.
  3. Species name to be mentioned in full while mentioning for the 1st time (right from abstract)
  4. A clear and concise methodology adopted to be included in abstract- lie how the authors determined the efficiency of pollinators.
  5. better sets of keywords to be used.
  6. 6.Introduction shall elaborate more on the decline of pollinators - causes, effects, decline rate in % and suggested conservation measures if any.
  7. 7.Figures are presented in a good manner - scientific names to be italicised.
  1. Tables could be formatted to be more understanding - shall add row borders within categories.
  2. Discussion - shall be rewritten - not proportionate with the amount of results obtained - each category of results shall be explained, justified with appropriate citations.

Author Response

Response to Reviewer 2 Comments

Dear reviewer,

On behalf of my co-authors, we thank you very much for giving us an opportunity to revise our manuscript (No. insects-1696754) again. We are very sorry for our incorrect in manuscript. We tried our best on revising contents of manuscript and replying reviewer’s question. We apologize again for our negligence.

We have not considered submitting other journals. It will be an honor for all our authors to publish in “Insects”. We have revised our manuscript according reviewers’ comments which we hope meet with approval. Special thanks to you for your patience and tolerance.

We will be happy to revise it again if our responses are not satisfied. Have a nice day.

Best regards!

Sincerely yours

Xiao-Ming Fan

On behalf of all authors.

------------------------------------------------------------------------------------------------------------------------------------

Point 1: Title needs to be changed to be more meaningful .

Response 1: Thank you for pointing out the problem with our title. We have compiled a title based on your comments and the comments of the other two reviewers, as follows:

What are the best pollinator candidates of Camelia oleifera: do not forget hoverflies and flies

Point 2: Importance of  f Camellia oleifera plant shall be added as a line in abstract - as scope or background of research.

Response 2: Thanks for pointing out our mistakes, I'm sorry for our mistakes. We have addressed them accordingly. The specific modifications are as follows:

Abstract: Camellia oleifera Abel is an important woody grain and oil plants, its pollination success is important for production. We conducted this study to select the best pollinator candidates of C. oleifera by principal component analysis and multi-attribute decision-making. Field observations of the flower-visiting characteristics of candidate pollinators were conducted at three sites. The insect species that visited flowers did not differ considerably between regions or time periods; however, the proportion of each species did vary. We recorded eleven main candidates from two orders and six families at the three sites. The pollen number carried on the of Apis mellifera was significantly higher than that of other insects; however, its visit frequency and body length were smaller than those of Vespa velutina. Statistical analysis shows that A. mellifera was the best candidate pollinator; Eristalis cerealis was a good candidate pollinator; Phytomia zonata, A. cerana, and V. velutina were ordinary candidate pollinators; and four fly species, Episyrphus balteatus, and E. arvorum were classified as inefficient candidate pollinators. Additionally, our study showed that flies and hoverflies also play an important role in the pollination system; hence, given the global decline in bee populations, they should also be considered in C. oleifera seed production.

Point 3: Species name to be mentioned in full while mentioning for the 1st time (right from abstract)

Response 3: Thank you for reporting our errors and we have addressed them accordingly.

The pollen number carried on the of Apis mellifera was significantly higher than that of other insects; however, its visit frequency and body length were smaller than those of Vespa velutina. Statistical analysis shows that A. mellifera was the best candidate pollinator; Eristalis cerealis was a good candidate pollinator; Phytomia zonata, A. cerana, and V. velutina were ordinary candidate pollinators; and four fly species, Episyrphus balteatus, and E. arvorum were classified as inefficient candidate pollinators. Additionally, our study showed that flies and hoverflies also play an important role in the pollination system; hence, given the global decline in bee populations, they should also be considered in C. oleifera seed production.

Point 4: A clear and concise methodology adopted to be included in abstract- lie how the authors determined the efficiency of pollinators.

Response 4: Thank you for pointing out our mistakes patiently. We've corrected them accordingly. Specific modifications include the following:

Abstract: Camellia oleifera Abel is an important woody grain and oil plants, its pollination success is important for production. We conducted this study to select the best pollinator candidates of C. oleifera by principal component analysis and multi-attribute decision-making. Field observations of the flower-visiting characteristics of candidate pollinators were conducted at three sites. The insect species that visited flowers did not differ considerably between regions or time periods; however, the proportion of each species did vary. We recorded eleven main candidates from two orders and six families at the three sites. The pollen number carried on the of Apis mellifera was significantly higher than that of other insects; however, its visit frequency and body length were smaller than those of Vespa velutina. Statistical analysis shows that A. mellifera was the best candidate pollinator; Eristalis cerealis was a good candidate pollinator; Phytomia zonata, A. cerana, and V. velutina were ordinary candidate pollinators; and four fly species, Episyrphus balteatus, and E. arvorum were classified as inefficient candidate pollinators. Additionally, our study showed that flies and hoverflies also play an important role in the pollination system; hence, given the global decline in bee populations, they should also be considered in C. oleifera seed production.

Point 5: better sets of keywords to be used.

Response 5: Thanks for your suggestion, and we've corrected them accordingly. Specific modifications include the following:

Keywords: Camellia oleifera Abel., pollinator candidates; Apis mellifera; hoverflies; flies

Point 6: Introduction shall elaborate more on the decline of pollinators - causes, effects, decline rate in % and suggested conservation measures if any.

Response 6: Thank you for your suggestion and we have addressed them accordingly.

however, the number and diversity of pollinators have declined globally owing to habitat fragmentation, pesticides use, climate change, and other factors. Back in the 1850s, 23 species of bees and wasps in Britain were told they were about to go extinct[7]. Subsequently, plant reproduction has also decreased, in cash crops, a decline in the variety and number of pollinators can reduce yields by 5% to 8%, even reaching levels close to extinction for some species. However, the level of extinction risk depends on the extent to which a species relies on pollinators for reproduction [8].

Point 7: Figures are presented in a good manner - scientific names to be italicised.

Response 7: Thank you for reporting our errors and we have addressed them accordingly.

Figure 1. Flower visitors of Camellia oleifera.

Figure 2. Posture of Camellia oleifera flower visitors. The red arrow represents the remaining pollen on the surface of the insect after taking it out of the penicillin bottle. Bar=1mm.

Figure 3. Cluster analysis of two methods. (a) Principal Component Analysis. (b) Multi-attribute decision-making.

Point 8: Tables could be formatted to be more understanding - shall add row borders within categories.

Response 8: Thank you for your suggestion and we have addressed them accordingly. The specific modifications are as follows:

Table 3. The proportion of Camellia oleifera flower visitors in different periods in site 1.

Categories

Family

Species

Proportion

First half of December

Second half of December

Volatility

Flies

Calliphoridae

Stomorhina obsoleta

12.14%

20.60%

8.46%

Lucilia sericata

5.00%

1.01%

-3.99%

Muscidae

Neomyia timorensis

4.29%

8.04%

3.75%

Sarcophagidae

Scrcophaganae morrhoidalis

11.43%

4.52%

-6.91%

Hoverflies

Syrphidae

Phytomia zonata

11.43%

15.08%

3.65%

Episyrphus balteatus

1.43%

10.05%

8.62%

Eristalis arvorum

8.57%

4.52%

-4.05%

Eristalis cerealis

2.86%

3.02%

0.16%

Bees

Apidae

Apis cerana

2.14%

4.02%

1.88%

Apis mellifera

3.57%

22.61%

19.04%

Wasps

Vespidae

Vespa velutina

37.14%

6.53%

-30.61%

Table 4. The main foraging behaviour of the main flower visitors of Camellia oleifera.

Categories

Species

Time for each visiting flower (s)

Single flower visit frequency (times/min)

Main foraging behaviour

The shortest

The longest

The lowest

The highest

Average frequency

Flies

Stomorhina obsoleta

9

96

1

5

2.17±1.24c

Spending more than half of their visiting anthers more, but touch the stigma less.

Lucilia sericata

16

22

2

4

3.00±0.77abc

Neomyia timorensis

8

160

1

5

2.25±1.16c

Scrcophaganae morrhoidalis

8

54

1

6

2.69±1.38bc

Hoverflies

Phytomia zonata

3

270

1

5

2.71±1.28bc

More active and more than half of them will touch the stigma.

Episyrphus balteatus

7

21

1

2

1.33±0.47c

Mainly inspect the flowers and stay for a short time.

Eristalis arvorum

5

78

1

6

2.71±1.28bc

Sometimes will rest on flowers temporarily without any activities.

Eristalis cerealis

5

34

1

3

2.67±0.60bc

Taking a short time to forage, but they will touch the stigma almost every time.

Bees

Apis cerana

3

46

1

3

2.33±0.58bc

Visiting time is short, and the enthusiasm of visiting flowers is low.

Apis mellifera

2

148

1

7

4.06±1.61ab

Visiting flower frequently, collecting pollen actively, and contacting stigma almost every time.

Wasps

Vespa velutina

2

127

1

6

4.47±1.27a

Visiting flower actively, and touch stigma almost every time.

Table 5. Pollen-carrying situation of main flower visitors in Camellia oleifera.

Categories

Species

Pollen load (grain/individual)

Main powder-carrying position

Total amount of pollen

Normal pollen

Pseudopollen

Pseudopollen / Normal pollen

Flies

Stomorhina obsoleta

2320±129.84d

1700±169.97c

620±220.10c

0.38±0.16ab

Back plate and head

Lucilia sericata

1600±1074.97d

1200±1032.80c

400±699.21c

0.43±0.79a

Neomyia timorensis

2000±608.58d

1833±593.17c

167±175.68c

0.10±0.11cd

Scrcophaganae morrhoidalis

3188±400.64d

2688±512.62c

500±238.37c

0.21±0.16abcd

Hoverflies

Phytomia zonata

12667±237.17c

11333±210.82b

1500±241.96b

0.12±0.09cd

Body surface and feet

Episyrphus balteatus

733±262.94d

600±262.94c

133±172.13c

0.30±0.42abc

Back plate and head

Eristalis arvorum

3533±688.53d

3400±733.67c

133±172.13c

0.04±0.06d

Body surface

Eristalis cerealis

48800±5391.35b

48200±5202.56a

667±699.21c

0.01±0.02d

Villi on the body surface

Bees

Apis cerana

14650±1106.80c

13300±948.68b

1350±411.64b

0.10±0.03cd

Pollen-carrying legs and villi

Apis mellifera

55167±6549.81a

47583±6120.09a

7500±707.11a

0.16±0.02bcd

Wasps

Vespa velutina

13000±4216.37c

11675±4323.79b

1325±373.61b

0.13±0.05cd

Villi on the body surface

Table 6. Posture characteristics of main flower-visiting insects in Camellia oleifera.

Categories

Species

Body length (mm)

Head width (mm)

Head length (mm)

Shoulder length (mm)

Shoulder width (mm)

Body surface characteristics

Flies

Stomorhina obsoleta

6.95±1.17f

2.28±0.51fg

2.01±0.43ef

2.47±0.58f

2.17±0.56h

Body surface bristles

Lucilia sericata

8.99±0.88e

3.44±0.33e

2.18±0.24def

3.62±0.37de

3.44±0.42cd

Neomyia timorensis

9.63±0.95de

3.58±0.24de

2.46±0.60cde

3.65±0.41de

3.25±0.26de

Scrcophaganae morrhoidalis

6.60±0.45f

2.19±0.35g

1.85±0.42f

2.66±0.21f

2.77±0.36fg

Bristled, sparse at the back

Hoverflies

Phytomia zonata

13.49±0.60b

4.99±0.18b

2.81±0.31c

4.82±0.41b

5.07±0.36b

Densely tomentose

Episyrphus balteatus

9.39±0.63e

2.57±0.19f

2.68±0.41cd

2.61±0.14f

2.43±0.10gh

Tomentose on both sides and short hairs on the ventral segment

Eristalis arvorum

10.63±1.06d

3.88±0.14cd

3.57±0.12b

3.98±0.27cd

3.77±0.38c

Dorsal plate is tomentose and the ventral segment is short-haired

Eristalis cerealis

12.58±1.65bc

4.15±0.91c

2.76±0.32c

4.12±0.56c

3.59±1.03cd

Densely tomentose, and the dorsal plate is particularly dense

Bees

Apis cerana

12.20±0.73c

3.66±0.18de

2.91±0.72c

3.70±0.31de

3.04±0.33ef

Densely covered with yellow villi, short ventral hairs, with pollen-carrying legs

Apis mellifera

13.05±0.65bc

3.45±0.24e

2.56±0.71cd

3.54±0.34e

2.97±0.57ef

Wasps

Vespa velutina

20.42±2.48a

5.34±0.26a

4.33±1.01a

6.06±0.47a

6.67±0.51a

Densely tomentose

Table 9. Principal component score, comprehensive score and ranking of main flower visitors in Camellia oleifera.

Categories

Species

Pollen load

Body surface characteristics

Body length

Proportion

Visiting frequency

PC1

PC2

F

Rank

Flies

Stomorhina obsoleta

-0.62

-0.84

-1.13

-0.37

-0.68

-0.93

0.47

-0.58

10

Lucilia sericata

-0.66

-0.84

-0.59

-0.43

0.27

-0.58

-0.20

-0.49

8

Neomyia timorensis

-0.64

-0.84

-0.43

-0.03

-0.59

-0.67

0.22

-0.45

6

Scrcophaganae morrhoidalis

-0.58

-0.84

-1.22

-0.46

-0.09

-0.81

0.30

-0.53

9

Hoverflies

Phytomia zonata

-0.09

0.70

0.58

-0.33

-0.06

0.22

-0.59

0.02

5

Episyrphus balteatus

-0.70

-0.84

-0.49

-0.05

-1.64

-0.97

0.57

-0.59

11

Eristalis arvorum

-0.56

-0.84

-0.17

-0.40

-0.06

-0.53

-0.30

-0.47

7

Eristalis cerealis

1.78

0.70

0.34

-0.23

-0.11

0.68

0.34

0.60

2

Bees

Apis cerana

0.02

1.47

0.24

-0.35

-0.50

0.27

-0.26

0.14

4

Apis mellifera

2.11

1.47

0.47

2.98

1.48

2.13

1.79

2.04

1

Wasps

Vespa velutina

-0.07

0.70

2.39

-0.34

1.99

1.18

-2.33

0.31

3

Table 11. Normalised score, comprehensive score and ranking of main flower visitors of Camellia oleifera.

Categories

Species

Pollen load

Body surface characteristics

Body length

Proportion

Visiting frequency

Total score

Rank

Flies

Stomorhina obsoleta

0.03

0.00

0.03

0.02

0.27

0.03

11

Lucilia sericata

0.02

0.00

0.17

0.01

0.53

0.07

8

Neomyia timorensis

0.02

0.00

0.22

0.12

0.29

0.07

7

Scrcophaganae morrhoidalis

0.05

0.00

0.00

0.00

0.43

0.05

9

Hoverflies

Phytomia zonata

0.22

0.67

0.50

0.04

0.44

0.38

5

Episyrphus balteatus

0.00

0.00

0.20

0.12

0.00

0.04

10

Eristalis arvorum

0.05

0.00

0.29

0.02

0.44

0.10

6

Eristalis cerealis

0.88

0.67

0.43

0.07

0.42

0.65

2

Bees

Apis cerana

0.26

1.00

0.41

0.03

0.32

0.46

4

Apis mellifera

1.00

1.00

0.47

1.00

0.86

0.91

1

Wasps

Vespa velutina

0.23

0.67

1.00

0.04

1.00

0.49

3

Point 9: Discussion - shall be rewritten - not proportionate with the amount of results obtained - each category of results shall be explained, justified with appropriate citations.

Response 9: Thank you for your question. We have compiled the discussion section based on your comments and the comments of the other two reviewers. In our revised Discussion, the first two paragraphs correspond to 3.1, 3.2, 3.3 of the Results section, i.e., the body and flower-visiting characteristics of insects; the latter two paragraphs discuss the selection of pollinator candidates.

  1. Discussion

Considering the body and flower-visiting characteristics of candidate pollinators, bees might be the best candidate pollinators; however, flies and hoverflies are also play an important role in C. oleifera pollination [26,27]. When land-use change, climate change, habitat fragmentation, and alien biological invasion occur, the diversity of insect pollinators in the ecosystem also changes. Insect pollinator communities can leverage functional redundancy to ensure the stability of an ecosystem via pollination function on a small spatial scale [28,29]. For example, when the chemical in C. oleifera nectar results in a reduction in bee numbers, other pollinators, such as flies and hoverflies, become the dominant pollinators and compensate for the absence of bees [30]. This dynamic regulation occurs because there is niche overlap among pollinator functional groups (species) [31]. This shows that an ecosystem with abundant pollinator functional groups can easily compensate for the loss of one or more groups [6].

Similar to previous studies[15], we found that unlike bees and wasps, which are highly motivated to visit flowers and touch the stigma almost every time they visit flowers, flies are small in size, have a low pollen load and they are not very motivated to visit flowers, often perching on the anthers without contacting the stigma, which makes it difficult for them to pollinate the plants. However, they regulate the stability of the pollination system as generalised pollinators. In some plant species, flies also act as specialised pollinators [32,33]. Unlike flies, hoverflies have a high pollen load and most individuals will contact the stigma when they eat nectar, thereby transferring the pollen to the stigma to participate in pollination.  Hoverflies can increase the crop yield to the same extent as bees. Furthermore, hoverflies have certain advantages over bees, such as the ability to carry pollen further, thus promoting the movement of pollen and flow of genes throughout the landscape [34,35]. Therefore, attention should be paid to the pollination efficiency of flies and hoverflies considering the gradual decline in bee populations [36].

We used statistical methods to select A. mellifera as the best C. oleifera candidate pollinator, and found that flies and hoverflies are also play a very important role in the pollination system. PCA and the MADM model showed similar results for the pollinator potential of flower visitors; however, the results of both analyses differed from those based on taxonomic affinities. This suggested that the classification of flower visitors into different functional groups should take into consideration the contribution of (different) flower visitors to pollination, while disregarding taxonomic affinities [37,38]. Like most flowering plants, C. oleifera has multiple functional visitor groups (flies, hoverflies, bees, and wasps); however, only a subset of these visitors serve as effective pollinators [39]. Most studies adhere to traditional or taxonomic groups to classify functional groups; nevertheless, even insects that are taxonomically related show many differences with respect to posture, visiting enthusiasm, distribution, and plant-pollinator interactions [38,40].

Additionally, we determined that E. cerealis is a ‘good’ candidate pollinator; P. zonata, A. cerana, and V. velutina are ‘ordinary’ candidate pollinators; and the remaining flower visitors were ‘inefficient’ candidate pollinators for C. oleifera. This division highlights the relative importance of diverse pollinators [38]. In this study, the most important indicator of the pollinator potential of flower visitors was the pollen load. However, unlike most plants, C. oleifera has pseudopollen, which is ineffective for pollination [41]. Therefore, we made a distinction between pollen and pseudopollen to better understand the effectiveness of the pollen carried by insects. The results showed that pseudopollen may not affect pollination by insects. Although insects carry pseudopollen during flower visits, the proportion of pseudopollen is approximately 0.1, which is lower than the 0.3 for pseudopollen to pollen on C. oleifera anthers [42]. This indicates that the collection of pseudopollen rather than pollen by insects may be selective; however, the criteria for this selection are not clear.

Reviewer 3 Report

Yuan et al. Pollination Efficiency of wild pollinators of Camellia oleifera: don’t forget hoverflies and flies

Dear Authors,

While I found the results of your paper interesting, some parts look poorly written. This makes the main message pretty hard to follow. Below I tried to suggest some improvements. However, I would recommend more serious polishing and professional editing.

Line 3: Change “don’t” to “do not.” This is a scientific manuscript!

Lines 15, 40: It is not a good style to start sentences with And, But, etc.

Lines 51-52: What are those substances? Are they natural or brought by pesticides or insecticides? Please, explain this to the reader in more detail!

Line 61: Replace “the pollinators species” with “pollinator species.”

Line 80: the first; the second

Line81: Change “5 sunny” to “five sunny.”

Line 82: the visits TO; not IN

Line 93: “1 min” change to “one min.”

Line 130: What do you mean by “The diversity of”? The diversity of insects? Remove “elastic.”

Line 132: Replace 2 and 6 with two and six.

Line 136: You mean that A. mellifera was the most numerous pollinator (A. mellifera was the largest one), or do you want to say that it was the largest insect there?

Lines 136-137: “Elastic” means “rubber,” or do you just want to report on changes observed across the time scale?

Lines 137-138: You mention “stage.” How do you define “the early stage” and “the late stage”? I don’t find these in the Methods. Please, specify this in lines 80-82, or use “First half of December” and “Second half of December” as done in Table 3.

Line 139: How much or many is “found in abundance?”

Line 142: I would replace “Different pollinators” with “Pollinators” or “Pollinator insects.”

Line 197: Replace hyphen (“-“) with a semicolon (“;”).

Lines 215-219: Delete these sentences as this text does not look like the discussion part.

Lines 220-231: These lines also don’t belong to the Discussion.

Line 232-234: The sentence: Regarding the pollination efficiency of the pollinators, PCA and MADM model showed similar results, but the results were different from those based on taxonomic affinities” is the first actual sentence of your Discussion. However, it doesn't look obvious to the reader. Please, start your Discussion by summarizing the main findings of your exciting study!

Line 234: You either show that bees are effective pollinators or show that they are not. Nobody is interested to know what you believe in this paragraph. Please, improve accordingly!

Author Response

Response to Reviewer 3 Comments

Dear reviewer,

On behalf of my co-authors, we thank you very much for giving us an opportunity to revise our manuscript (No. insects-1696754) again. We are very sorry for our incorrect in manuscript. We tried our best on revising contents of manuscript and replying reviewer’s question. We apologize again for our negligence.

We have not considered submitting other journals. It will be an honor for all our authors to publish in “Insects”. We have revised our manuscript according reviewers’ comments which we hope meet with approval. Special thanks to you for your patience and tolerance.

We will be happy to revise it again if our responses are not satisfied. Have a nice day.

Best regards!

Sincerely yours

Xiao-Ming Fan

On behalf of all authors.

------------------------------------------------------------------------------------------------------------------------------------

Point 1: Line 3: Change “don’t” to “do not.” This is a scientific manuscript!

Response 1: Thank you for reporting our errors and we have addressed them accordingly. The specific modifications are as follows:

What are the best pollinator candidates of Camelia oleifera: do not forget hoverflies and flies

Point 2: Lines 15, 40: It is not a good style to start sentences with And, But, etc.

Response 2: Thanks for pointing out our grammatical mistakes, I'm sorry for our mistakes. We have addressed them accordingly. The specific modifications are as follows:

We found that Apis mellifera was the best candidate pollinator, but flies and hoverflies also played an important role in the pollination system.

However, the level of extinction risk depends on the extent to which a species relies on pollinators for reproduction

Point 3: Lines 51-52: What are those substances? Are they natural or brought by pesticides or insecticides? Please, explain this to the reader in more detail!

Response 3: Thank you for your suggestion, we have added it to the relevant position in the introduction, as follows:

However, managed honeybee populations placed in C. oleifera forests have a high death rate because C. oleifera nectar contains strong alkaloids and other indigestible compounds that cause posterior intestinal obstruction in bee larvae [15,16].

Point 4: Line 61: Replace “the pollinators species” with “pollinator species.”

Response 4: Thank you for reading our article carefully and patiently pointing out mistakes for us. We've corrected them accordingly. Specific modifications include the following:

The proportion of each insect species, flower visit duration and frequency, and other indices were quantified, and the body and flower-visiting characteristics of the insects were analysed and compared.

Point 5: Line 80: the first; the second

Response 5: Thanks for pointing out our grammatical mistakes, and we've corrected them accordingly. Specific modifications include the following:

To investigate the flower visitors visits to C. oleifera during the flowering period (from November to January), we conducted the following experiments in December (the temperature in the first half of the month was 6–16 ℃ and in the second half was 3–11 ℃). We observed 2679 insects in the process. (a) We chose five sunny days in the first half of December at three sites and observed the flower-visiting insects from 11:00 a.m.–12:00pm., when visitors are most active [18]; (b) we observed the visits to each site in the second half of December. The proportions of visiting insect species were calculated using the following formula:

Insect proportion = Single species insect number/total insect number

Point 6: Line81: Change “5 sunny” to “five sunny.”

Response 6: Thank you for your suggestion and we have addressed them accordingly.

We chose five sunny days in the first half of December at three sites and observed the flower-visiting insects from 11:00 a.m.–12:00pm., when visitors are most active [18].

Point 7: Line 82: the visits TO; not IN

Response 7: Thank you for reporting our errors and we have addressed them accordingly.

we observed the visits to each site in the second half of December. The proportions of visiting insect species were calculated using the following formula:

Point 8: Line 93: “1 min” change to “one min.”

Response 8: Thank you for your suggestion and we have addressed them accordingly. The specific modifications are as follows:

visiting frequency, i.e. number of insect visits to the flowers in one min

Point 9: Line 130: What do you mean by “The diversity of”? The diversity of insects? Remove “elastic.”

Response 9: Thank you for your question. "Diversity" means a diversity of insects.

Point 10: Line 132: Replace 2 and 6 with two and six.

Response 10: Thank you for reporting our errors and we have addressed them accordingly. The specific modifications are as follows:

Twelve species of insect frequency appeared and contacted the stigmas in the three sites (Fig. 1). They came from two orders and six families. The proportions of species at the three sites were different, e.g. V. mandarinia was not found at site 1 (Table 2).

Point 11: Line 136: You mean that A. mellifera was the most numerous pollinator (A. mellifera was the largest one), or do you want to say that it was the largest insect there?

Response 11: I'm sorry, our language is skewed, and we've revised it accordingly. The specific modifications are as follows:

At site 1, V. velutina was the largest proportion of species, but at sites 2 and 3, A. mellifera has the largest proportion.

Point 12: Lines 136-137: “Elastic” means “rubber,” or do you just want to report on changes observed across the time scale?

Response 12: Thank you for your question. Here, we simply want to express the change in the proportion of insects in time.

Point 13: Lines 137-138: You mention “stage.” How do you define “the early stage” and “the late stage”? I don’t find these in the Methods. Please, specify this in lines 80-82, or use “First half of December” and “Second half of December” as done in Table 3.

Response 13: Thank you for your suggestion. The “early stage” is the first half of December, and the later is the second half of December. To better express this viewpoint, we have made the following changes to the content of the article:

In the first half of December, wasps were the main insects, but in the second half of December, the proportions of wasps and bees fluctuated greatly. Flies and hoverflies were found in a large proportion (Table 3).

Point 14: Line 139: How much or many is “found in abundance?”

Response 14: We apologize for the vague phrase here, and revised in the article:

the proportions of wasps and bees fluctuated greatly. Flies and hoverflies were found in a large proportion

Point 15: Line 142: I would replace “Different pollinators” with “Pollinators” or “Pollinator insects.”

Response 15: Thank you for your suggestion. By combining your comments with those of the first reviewer, we have implemented the following changes to the content of the article:

Figure 1. Flower visitors of Camellia oleifera.

Point 16: Line 197: Replace hyphen (“-“) with a semicolon (“;”).

Response 16: Thank you for reporting our errors and we have addressed them accordingly. The specific modifications are as follows:

Thus, we obtained the scores for each insect species; the score of A. mellifera was much higher than those of other insects, reaching 0.91 (Table 11).

Point 17: Lines 215-219: Delete these sentences as this text does not look like the discussion part.

Response 17: Thank you for pointing out our mistakes. We have deleted this section.

Point 18: Lines 220-231: These lines also don’t belong to the Discussion.

Response 18: Thank you for reporting our errors and we have addressed them accordingly. The specific modifications are as follows:

  1. Discussion

 We used statistical methods to select A. mellifera as the best C. oleifera candidate pollinator, and found that flies and hoverflies are also play a very important role in the pollination system. PCA and the MADM model showed similar results for the pollinator potential of flower visitors; however, the results of both analyses differed from those based on taxonomic affinities. This suggested that the classification of flower visitors into different functional groups should take into consideration the contribution of (different) flower visitors to pollination, while disregarding taxonomic affinities [37,38]. Like most flowering plants, C. oleifera has multiple functional visitor groups (flies, hoverflies, bees, and wasps); however, only a subset of these visitors serve as effective pollinators [39]. Most studies adhere to traditional or taxonomic groups to classify functional groups; nevertheless, even insects that are taxonomically related show many differences with respect to posture, visiting enthusiasm, distribution, and plant-pollinator interactions [38,40].

Point 19: Line 232-234: The sentence: Regarding the pollination efficiency of the pollinators, PCA and MADM model showed similar results, but the results were different from those based on taxonomic affinities” is the first actual sentence of your Discussion. However, it doesn't look obvious to the reader. Please, start your Discussion by summarizing the main findings of your exciting study!

Response 19: Thank you for your suggestion and we have addressed them accordingly. The specific modifications are as follows:

We used statistical methods to select A. mellifera as the best C. oleifera candidate pollinator, and found that flies and hoverflies are also play a very important role in the pollination system. PCA and the MADM model showed similar results for the pollinator potential of flower visitors; however, the results of both analyses differed from those based on taxonomic affinities.

Point 20: Line 234: You either show that bees are effective pollinators or show that they are not. Nobody is interested to know what you believe in this paragraph. Please, improve accordingly!

Response 20: Thank you for your suggestion and we have addressed them accordingly. The specific modifications are as follows:

We used statistical methods to select A. mellifera as the best C. oleifera candidate pollinator, and found that flies and hoverflies are also play a very important role in the pollination system. PCA and the MADM model showed similar results for the pollinator potential of flower visitors; however, the results of both analyses differed from those based on taxonomic affinities. This suggested that the classification of flower visitors into different functional groups should take into consideration the contribution of (different) flower visitors to pollination, while disregarding taxonomic affinities [37,38]. Like most flowering plants, C. oleifera has multiple functional visitor groups (flies, hoverflies, bees, and wasps); however, only a subset of these visitors serve as effective pollinators [39]. Most studies adhere to traditional or taxonomic groups to classify functional groups; nevertheless, even insects that are taxonomically related show many differences with respect to posture, visiting enthusiasm, distribution, and plant-pollinator interactions [38,40].

Additionally, we determined that E. cerealis is a ‘good’ candidate pollinator; P. zonata, A. cerana, and V. velutina are ‘ordinary’ candidate pollinators; and the remaining flower visitors were ‘inefficient’ candidate pollinators for C. oleifera. This division highlights the relative importance of diverse pollinators [38]. In this study, the most important indicator of the pollinator potential of flower visitors was the pollen load. However, unlike most plants, C. oleifera has pseudopollen, which is ineffective for pollination [41]. Therefore, we made a distinction between pollen and pseudopollen to better understand the effectiveness of the pollen carried by insects. The results showed that pseudopollen may not affect pollination by insects. Although insects carry pseudopollen during flower visits, the proportion of pseudopollen is approximately 0.1, which is lower than the 0.3 for pseudopollen to pollen on C. oleifera anthers [42]. This indicates that the collection of pseudopollen rather than pollen by insects may be selective; however, the criteria for this selection are not clear.

Round 2

Reviewer 1 Report

The authors made a significant improvement in the manuscript. The paper is smoother to read, and additional explanations and data analyses are welcome. They provided insect behavior analyses that will be valuable for numerous studies, with a larger scoop than the single plant species observed in this study.

I have two small remarks:

L.4 "Yuan a,b* " I have the impression it should be "1,2" instead of "a,b"

Fig 1. Are these data means over all the sites? Please can you specify if the figures are sum, mean, etc., and the number of sites pooled together. Detailing on the graph what is density (insect visits/600 flowers/h) would make it more straightforward.

Author Response

Response to Reviewer 1 Comments

Dear reviewer,

On behalf of my co-authors, we thank you very much for giving us an opportunity to revise our manuscript (No. insects-1696754) again. We are very sorry for our incorrect in manuscript. We tried our best on revising the content of the manuscript and replying reviewer’s question. We apologize again for our negligence.

We have not considered submitting other journals. It will be an honor for all our authors to publish in “Insects”. We have revised our manuscript according reviewers’ comments which we hope to meet with approval. Special thanks to you for your patience and tolerance.

We will be happy to revise it again if our responses are not satisfied. Have a nice day.

Best regards!

Sincerely yours

Xiao-Ming Fan

On behalf of all authors.

------------------------------------------------------------------------------------------------------------------------------------

Point 1: L.4 "Yuan a,b* " I have the impression it should be "1,2" instead of "a,b"

Response 1: Thank you for pointing out our mistake, which we have revised. The specific modifications are as follows:

Bin Yuan 1,2,†, Guan-xing Hu 1,2,†, Xiao-xiao Zhang 1,2, Jing-kun Yuan 1,2, Xiao-ming Fan 1,2,* and De-yi Yuan 1,2*

Point 2: Fig 1. Are these data means over all the sites? Please can you specify if the figures are sum, mean, etc., and the number of sites pooled together. Detailing on the graph what is density (insect visits/600 flowers/h) would make it more straightforward.

Response 2: I'm sorry, we didn't express our meaning clearly in the manuscript. We mixed the data from three sits, and on the same day, we conducted a survey on the flower-visiting density at three sits at the same time, finally we mixed the data from three sits in the same period. Take the average as a recorded data. With regard to the figure, we have revised it in accordance with your suggestion. Once again, we thank you for your comments. The specific amendments are as follows:

2.3 Flower-visiting density

To select the best observation period for the flower-visiting insects, the flower-visiting density of C. oleifera was recorded and classified from 8:00 to 18:00 on three sunny days when the shrubs were in full bloom. The statistical method for examining the flower visiting density is as follows: Three rows of plants were randomly selected for each sampling session at the site, and then banded sampling was carried out. The number of insects landing on 200 flowers in the row was recorded back and forth during one hour, and the data from the three rows were added together to represent the overall flower visiting density for the site. This experiment was carried out in three sites at the same time. After getting the data of each site, the average flower-visiting density at different periods of the three sites was recorded as a data.

Figure 1. Density of pollinators visiting the flowers of Camellia oleifera.

Reviewer 2 Report

The manuscript can be accepted now

Author Response

Response to Reviewer 2 Comments

Dear reviewer,

Thanks very much for your kind work and consideration on publication of our paper(No. insects-1696754). On behalf of my co-authors, we would like to express our great appreciation to you.

Best regards!

Sincerely yours

Xiao-Ming Fan

On behalf of all authors.

----------------------------------------------------------------------------------------

Reviewer 3 Report

Dear Authors,

I have just a couple of minor comments left:

Lines 17-18: Double “important” in one sentence. I would replace the second one with “essential”.

Line 17: “oil plant”, not “plants”

Line 23: Delete “the of”

Line 40: “pesticide use”, not “pesticides”

Line 42: “decreased; in cash crops

Line 50: Replace “which makes” with “making”

Line 85: Replace “the process” with “this study”

Line 86: 12:00 p.m.

Lines 100 and 101: I suggest replacing “in the process” with “in total”

Line 112: Data analyses

Line 146: balteatus, who

Line 147: visited to collect nectar

Line 162: Flies spent

Line 229: are also playing

Line 250: the flow of genes

Line 252: hoverflies, considering

Line 254: Change “a very important role” to “a crucial role”

Line 258: Change “take into consideration” to “consider”

Line 263: even insects that are taxonomically related taxonomically related insects

Line 271: Change “made a distinction” to “distinguished”

Line 271: to understand better

Author Response

Response to Reviewer 3 Comments

Dear reviewer,

On behalf of my co-authors, we thank you very much for giving us an opportunity to revise our manuscript (No. insects-1696754) again. We are very sorry for our incorrect in manuscript. We tried our best on revising contents of manuscript and replying reviewer’s question. We apologize again for our negligence.

We have not considered submitting other journals. It will be an honor for all our authors to publish in “Insects”. We have revised our manuscript according reviewers’ comments which we hope meet with approval. Special thanks to you for your patience and tolerance.

We will be happy to revise it again if our responses are not satisfied. Have a nice day.

Best regards!

Sincerely yours

Xiao-Ming Fan

On behalf of all authors.

------------------------------------------------------------------------------------------------------------------------------------

Point 1: Lines 17-18: Double “important” in one sentence. I would replace the second one with “essential”.

Response 1: Thank you for reporting our errors and we have addressed them accordingly. The specific modifications are as follows:

Camellia oleifera Abel is an important woody grain and oil plant, its pollination success is essential for production.

Point 2: Line 17: “oil plant”, not “plants”

Response 2: Thanks for pointing out our mistakes, I'm sorry for our mistakes. We have addressed them accordingly. The specific modifications are as follows:

Camellia oleifera Abel is an important woody grain and oil plant, its pollination success is essen-tial for production.

Point 3: Line 23: Delete “the of”

Response 3: Thank you for your suggestion, we have added it to the relevant position in the introduction, as follows:

The pollen number carried on Apis mellifera was significantly higher than that of other insects

Point 4: Line 40: “pesticide use”, not “pesticides”

Response 4: Thank you for reading our article carefully and patiently pointing out mistakes for us. We've corrected them accordingly. Specific modifications include the following:

however, the number and diversity of pollinators have declined globally owing to habitat fragmentation, pesticide use, climate change, and other factors.

Point 5: Line 42: “decreased; in cash crops

Response 5: Thanks for pointing out our mistakes, and we've corrected them accordingly. Specific modifications include the following:

Subsequently, plant reproduction has also decreasedï¼›in cash crops

Point 6: Line 50: Replace “which makes” with “making”

Response 6: Thank you for your suggestion and we have addressed them accordingly.

Insects are the most important pollinators, contributing to 87% of the pollination worldwide, which making their services critical to the sustainability of natural ecosystems

Point 7: Line 85: Replace “the process” with “this study”

Response 7: Thank you for your suggestion and we have addressed them accordingly.

We observed 2679 insects in this study.

Point 8: Line 86: 12:00 p.m.

Response 8: Thank you for your suggestion and we have addressed them accordingly. The specific modifications are as follows:

We chose five sunny days in the first half of December at three sites and observed the flower-visiting insects from 11:00 a.m.–12:00 p.m., when visitors are most active

Point 9: Lines 100 and 101: I suggest replacing “in the process” with “in total”

Response 9: Thank you for your suggestion and we have addressed them accordingly. The specific modifications are as follows:

visiting frequency, i.e. number of insect visits to the flowers in one min, we recorded 168 insects in total

Point 10: Line 112: Data analyses

Response 10: Thank you for reporting our errors and we have addressed them accordingly. The specific modifications are as follows:

2.7 Data analyses

Point 11: Line 146: balteatus, who

Response 11: I'm sorry, our language is skewed, and we've revised it accordingly. The specific modifications are as follows:

except Episyrphus balteatus, who often hovered over but did not visit the flowers.

Point 12: Line 147: visited to collect nectar

Response 12: I'm sorry, our language is skewed, and we've revised it accordingly. The specific modifications are as follows:

Most of the bees visited to collect nectar

Point 13: Line 162: Flies spent

Response 13: I'm sorry, our language is skewed, and we've revised it accordingly. The specific modifications are as follows:

Flies spent more than half of their time visiting anthers and inhabited the anther in-stead of the stigma.

Point 14: Line 229: are also playing

Response 14: Thank you for your suggestion. We have made the following changes to the content of the article:

, flies and hoverflies are also playing an important role in C. oleifera pollination

Point 14: Line 250: the flow of genes

Response 14: Thanks for pointing out our mistakes, and we've corrected them accordingly. Specific modifications include the following:

thus promoting the movement of pollen and the flow of genes throughout the landscape

Point 15: Line 252: hoverflies, considering

Response 15: Thanks for pointing out our mistakes, and we've corrected them accordingly. Specific modifications include the following:

Therefore, attention should be paid to the pollination efficiency of flies and hoverflies, considering the gradual decline in bee populations

Point 16: Line 254: Change “a very important role” to “a crucial role”

Response 16: I'm sorry, our language is skewed, and we've revised it accordingly. The specific modifications are as follows:

and found that flies and hoverflies are also play a crucial role in the pollination system.

Point 17: Line 258: Change “take into consideration” to “consider”

Response 17: Thank you for pointing out our mistakes.

This suggested that the classification of flower visitors into different functional groups should consider the contribution of (different) flower visitors to pollination

Point 18: Line 263: even insects that are taxonomically related taxonomically related insects

Response 18: Thank you for reporting our errors and we have addressed them accordingly. The specific modifications are as follows:

Most studies adhere to traditional or taxonomic groups to classify functional groups; nevertheless, even taxonomically related insects show many differences with respect to posture

Point 19: Line 271: Change “made a distinction” to “distinguished”

Response 19: Thank you for your suggestion and we have addressed them accordingly. The specific modifications are as follows:

Therefore, we distinguished between pollen and pseudopollen to understand better the effectiveness of the pollen carried by insects.

Point 20: Line 271: to understand better

Response 20: Thank you for your suggestion and we have addressed them accordingly. The specific modifications are as follows:

Therefore, we distinguished between pollen and pseudopollen to understand better the effectiveness of the pollen carried by insects.

Round 3

Reviewer 3 Report

All my comments have been addressed. Thanks!

Author Response

Response to Reviewer 3 Comments

Dear reviewer,

Thanks very much for your kind work and consideration on publication of our paper(No. insects-1696754). On behalf of my co-authors, we would like to express our great appreciation to you.

Best regards!

Sincerely yours

Xiao-Ming Fan

On behalf of all authors.

----------------------------------------------------------------------------------------

This manuscript is a resubmission of an earlier submission. The following is a list of the peer review reports and author responses from that submission.